# LightMem: Lightweight and Efficient Memory-Augmented Generation

**Jizhan Fang♠, Xinle Deng♠, Haoming Xu♠, Ziyan Jiang♠, Yuqi Tang♠, Ziwen Xu♠,**
**Shumin Deng◇, Yunzhi Yao♠, Mengru Wang♠, Shuofei Qiao♠, Huajun Chen♠, Ningyu Zhang♠♣***
♠Zhejiang University    ◇National University of Singapore
♣State Key Lab. for Novel Software Technology, Nanjing University, P.R. China
{fangjizhan, zhangningyu}@zju.edu.cn

## Abstract

Despite their remarkable capabilities, Large Language Models (LLMs) struggle to effectively leverage historical interaction information in dynamic and complex environments. Memory systems enable LLMs to move beyond stateless interactions by introducing persistent information storage, retrieval, and utilization mechanisms. However, existing memory systems often introduce substantial time and computational overhead. To this end, we introduce a new memory system called **LightMem**, which strikes a balance between the performance and efficiency of memory systems. Inspired by the Atkinson–Shiffrin model of human memory, **LightMem** organizes memory into three complementary stages. First, cognition-inspired sensory memory rapidly filters irrelevant information through lightweight compression and groups information according to their topics. Next, topic-aware short-term memory consolidates these topic-based groups, organizing and summarizing content for more structured access. Finally, long-term memory with sleep-time update employs an offline procedure that decouples consolidation from online inference. On LONGMEMEVAL and LoCoMo, using GPT and Qwen backbones, **LightMem** consistently surpasses strong baselines, improving QA accuracy by up to 7.7% / 29.3%, reducing total token usage by up to 38× / 20.9× and API calls by up to 30× / 55.5×, while purely online test-time costs are even lower, achieving up to 106× / 117× token reduction and 159× / 310× fewer API calls. The code is available at https://github.com/zjunlp/LightMem.

## 1 Introduction

Memory is fundamental to intelligent agent, enabling the assimilation of prior experiences, contextual cues, and task-specific knowledge that underpin robust reasoning and decision-making (Wang et al., 2024; Behrouz et al., 2024; Du et al., 2025; Zhang et al., 2024). While Large Language Models (LLMs) (DeepSeek-AI et al., 2025; Achiam et al., 2023) demonstrate remarkable capabilities across a wide range of tasks, they exhibit significant limitations when engaged in long-context or multi-turn interaction scenarios due to fixed context windows and the "lost in the middle" problem (Liu et al., 2024). Memory systems are pivotal for overcoming these limitations, as they allow LLMs to maintain a persistent state across extended interactions. Recent works (Li et al., 2025b; Yang et al., 2024; Chhikara et al., 2025; Kang et al., 2025) address this challenge by building explicit external memory through sequential summarization and long term storage, enabling models to retain and retrieve relevant information over long horizons.

Note that a typical LLM memory system processes raw interaction data into manageable chunks, such as turn- or session-level in dialogue scenarios (Xu et al., 2025; Li et al., 2025a), organizes them into long-term memory (e.g., databases or knowledge graphs) by indexing them into memory units, and continuously updates by adding new information and discarding outdated or conflicting content (Zhong et al., 2024). This enables retrieval of relevant memories, improving coherence, and personalization in long-context, multi-turn scenarios.

---

*Corresponding Author.

**Challenges.** Despite these advances, as shown in Figure 1, contemporary memory systems still suffer from significant inefficiencies and consistency issues. First, in long interactions (e.g., dialogue scenarios), both user inputs and model responses often contain substantial redundant information (Maharana et al., 2024; Wu et al., 2025). Such information is typically irrelevant to downstream tasks or subsequent memory construction, and in some cases, may even negatively affect the model's in-context learning capability (Liu et al., 2023; Pan et al., 2025). However, current mainstream memory-related studies generally process the raw information directly without any filtering or refinement, leading to high overhead from noisy or irrelevant data. This inflates token consumption without proportional gains in reasoning quality or coherence. Second, memory construction typically **treats each turn in isolation or relies on rigid context-window boundaries**, failing to model semantic connections across different turns (Tan et al., 2025). As a result, during subsequent memory item construction, the backbone LLM may generate inaccurate or incomplete item representations due to overly entangled topics or semantics, leading to the loss of crucial contextual details. Third, memory updates and forgetting are usually performed directly **during inference and task execution**. This tight coupling introduces long test-time latency in long-horizon tasks and prevents deeper, reflective processing of past experiences.

**Building Lightweight Memory.** Inspired by the efficiency and structure of human memory, we introduce **LightMem**. In particular, LightMem emulates human memory through three key components: (1) A *pre-compression sensory memory module* that filters redundant or low-value tokens from raw input and buffers the distilled content. (2) A *topic-aware short-term memory* that leverages semantic and topical similarity to dynamically group related utterances into coherent segments. By adaptively determining segment boundaries based on content instead of fixed window sizes, this module produces more concentrated and meaningful memory units. This not only reduces the frequency of memory construction but also enables more

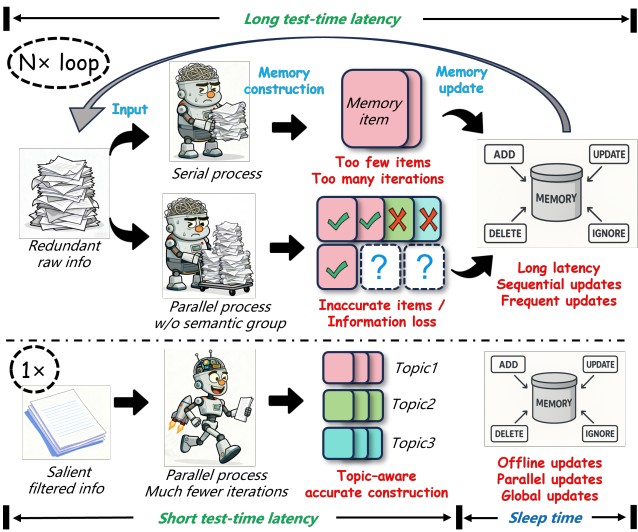

Figure 1: Comparison of previous works and **LightMem**.

precise and efficient retrieval during inference. (3) A *sleep-time update* mechanism for long-term memory maintenance. New memory entries are initially stored with timestamps to support immediate ("soft") updates for real-time responsiveness. Later, during designated offline periods (i.e., "sleep"), the system reorganizes, de-duplicates, and abstracts these entries, resolving inconsistencies and strengthening cross-knowledge connections. Crucially, this decouples expensive memory maintenance from real-time inference, enabling reflective, high-fidelity updates without introducing latency. By systematically filtering, organizing, and consolidating relevant information, LightMem substantially reduces computational overhead and API costs while sustaining accurate, coherent reasoning over extended interactions. We detail each component in §3.

**Results and Evaluation.** On LongMemEval (Wu et al., 2025), LightMem consistently outperforms the strongest baseline, improving accuracy by 2.09%–6.40% with GPT and up to 7.67% with Qwen. In terms of overall efficiency (online + offline), LightMem reduces total token usage by up to 38× for GPT and 21.8× for Qwen, lowers API calls by up to 30× and 17.1×, and accelerates runtime by up to 12.4× and 6.3×, respectively. If considering only online test-time costs, the gains become even larger: LightMem cuts token usage by up to 105.9× (GPT) and 117.1× (Qwen), and reduces API calls by up to 159.4× and 309.9×. On the LoCoMo benchmark (Maharana et al., 2024), LightMem maintains strong advantages, achieving 6.10%–29.29% higher accuracy and substantial efficiency improvements—boosting token efficiency by up to 20.92×, reducing API calls by up to 55.48×, and speeding up runtime by up to 8.21× across GPT and Qwen backbones. Furthermore, case studies in §5.6 show that the offline "sleep-time" consolidation enhances long-term memory reliability, mitigating information loss.

## 2 PRELIMINARY

### 2.1 CONVENTIONAL MEMORY SYSTEMS FOR LLMS

We describe mainstream memory architectures pipeline in terms of two major stages. **(I) Memory Bank Construction.** This stage can be further decomposed into three sub-stages: (a) Raw data $D$ are first processed at a chosen level of granularity, $D^{(g)} = f_{\text{seg}}(D; g), g \in \{\text{turn}, \text{session}, \text{topic}\}$ in dialog scenario; (b) The segmented data $D^{(g)}$ are then summarized or extracted to generate memory entries, $E = f_{\text{sum}}(D^{(g)})$, which are stored and organized within structural backends such as vector databases or knowledge graphs to enable long-term retention; (c) Many systems incorporate an updating mechanism to mitigate issues such as context conflicts or outdated information, $M' = f_{\text{update}}(M, R; U)$, where $M$ denotes the existing memory bank, $R$ represents newly generated memory entries, and $U$ specifies the update or forgetting policy. **(II) Retrieval and Usage.** When a new user query arrives, the system retrieves relevant entries from the memory bank, integrates them with the query to construct the final prompt, and then invokes the model to produce a response.

### 2.2 LIMITATIONS OF EXISTING LLM MEMORY SYSTEMS

Compared to human memory, current LLM memory systems are burdened by high maintenance costs, mainly due to three limitations: **1) Redundant Sensory Memory.** In current systems, $f_{\text{sum}}()$ and $f_{\text{gran}}(; g = \text{topic})$ are typically executed by calling stronger LLMs. Feeding raw data $D$ directly wastes resources and even weakens in-context learning due to redundancy. A key challenge is to design lightweight mechanisms that pre-compress inputs and apply pre-attention strategies to capture semantic units at different granularities efficiently. **2) Balancing Effectiveness and Efficiency in STM.** As shown in Figure 1, when input granularity is fixed, $D^{(g)}$ must pass through the entire pipeline. Excessively fine granularity increases latency and underutilizes STM capacity, whereas overly coarse granularity without semantic constraints or grouping may cause mixed or entangled semantics and topics, leading to inaccurate memory construction and loss of fine-grained details in subsequent processes. This calls for strategies that better balance effectiveness and efficiency in STM. **3) Inefficient LTM Updating.** Current $f_{\text{update}}()$ mechanisms face two main issues: (i) enforcing strict real-time updates at test time incurs significant latency, whereas STM can provide short-term context without immediate LTM updates; (ii) memory banks are updated sequentially due to ordering constraints (read-after-write/write-after-read), rather than being triggered dynamically. These limitations raise a research question: *Can we design LLM memory that is both efficient and lightweight, inspired by human memory mechanisms?*

## 3 LIGHTMEM ARCHITECTURE

Analogous to the human memory, we design LightMem as shown in Figure 2, which consists of three light modules: *Light1* implements an efficient *Sensory Memory Module* that selectively preserves salient information from raw input (§3.1), *Light2* realizes a topic-aware *STM Module* for transient information processing (§3.2), and *Light3* provides an *LTM module* designed to minimize test time update latency (§3.3) with a sleep time update mechanism. The overall pipeline framework of **LightMem**, its specific models, and comparisons with other memory frameworks are presented in Appendix A.1. The complexity analysis for LightMem's efficiency gains is in Section 4.

### 3.1 LIGHT1: COGNITIVE-INSPIRED SENSORY MEMORY

In long horizon interaction scenarios, such as user–assistant dialogues, a large portion of the information is redundant. Therefore, we design a *Pre-Compressing Submodule* to eliminate redundant tokens, followed by the *Topic Segmentation Submodule* that forms semantic topic-based segments for following faster and more accurate memory construction.

**Pre-Compressing Submodule.** This module leverages a compression model $\theta$ to eliminate redundant tokens, tailored for compatibility with the downstream memory construction phase:

$$\hat{\mathbf{x}} = \{x_i \in \mathbf{x} \mid P(\text{retain } x_i \mid \mathbf{x}; \theta) > \tau\}, \tau = \text{Percentile}(\{x_j\}, r),$$

Following Xia et al. (2025), we use LLMLingua-2 (Pan et al., 2024b) as our compression model $\theta$. Let $\mathbf{x}$ be the raw input tokens, $\theta$ the model, and $r$ the compression ratio. The threshold $\tau$ is set to

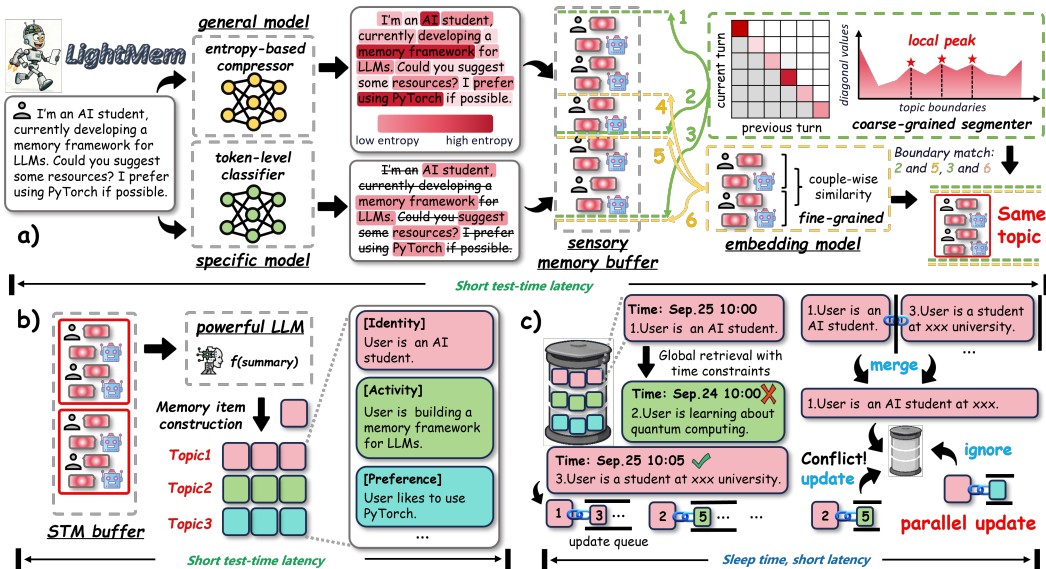

Figure 2: The **LightMem** architecture. **LightMem** consists of three modules: *a)* An efficient *Sensory Memory Module*, *b)* a topic aware *STM Module*, and *c)* an *LTM module* updated in sleep time.

the $r$-th percentile of retention scores, keeping only tokens above $\tau$. For $P(\text{retain } x_i \mid \mathbf{x})$, we treat the compression process as a binary token classification task ("retain" or "discard"). For each token $x_i$ in a sequence $\mathbf{x}$, the model $\theta$ outputs a logit vector $\ell_i$, and the retention probability is given by:

$$P(\text{retain } x_i \mid \mathbf{x}; \theta) = \text{softmax}(\ell_i)_1,$$

where the subscript 1 denotes the "retain" class. Tokens with probabilities above a dynamic threshold are included in the compressed sequence. In addition, **LightMem** can also employ more general generative LLM as the pre-compression model. We further implement a token filtering mechanism based on the cross-entropy between the model's predicted distribution and the true token labels:

$$P(\text{retain } x_i \mid \mathbf{x}; \theta) = -\sum_{x_i \in \mathcal{V}} q(x_i) \log P(x_i \mid \mathbf{x}; \theta)$$

where $q(x_i)$ denotes the true token label distribution. Tokens with higher conditional entropy under a given context are more uncertain and less predictable, indicating greater informational uniqueness and a more critical role in semantic expression, such distinctive tokens are essential for subsequent memory construction and are therefore retained.

**Topic Segmentation Submodule.** Existing works indicate that topic-granular input facilitates improved performance in memory systems (Pan et al., 2025; Tan et al., 2025). As shown in Figure 2, **LightMem** maintains a sensory memory buffer to temporarily store information after pre-compression. When the accumulated information reaches the buffer's maximum capacity, a hybrid topic segmentation operation based on attention and similarity is triggered. We use the compression model $\theta$ and an embedding model to compute attention matrices and semantic similarities, respectively. We define the final segmentation boundaries as the intersection of attention-based boundaries $\mathcal{B}_1$ and similarity-based boundaries $\mathcal{B}_2$:

$$\mathcal{B}_1 = \left\{ k \mid M_{k,k-1} > M_{k-1,k-2}, \ M_{k,k-1} > M_{k+1,k}, \ 1 < k < n \right\},$$

$$\mathcal{B}_2 = \left\{ k \mid \text{sim}(s_{k-1}, s_k) < \tau, 1 \leq k < n \right\}, \quad \mathcal{B} = \mathcal{B}_1 \cap \mathcal{B}_2.$$

Specifically, dialogue scenarios possess natural semantic units, namely the conversational turn. We construct a turn-level attention matrix $M \in \mathbb{R}^{n \times n}$. $\mathcal{B}_1$ are identified as local maxima in the sequence $\{M_{k,k-1}\}$, i.e., the sub-diagonal elements of $M$ corresponding to attention between consecutive sentences. The detailed process of $\mathcal{B}_1$ and illustrative cases are provided in Appendix C.1. To mitigate

attention sinks and dilution in attention-based methods, we compute semantic similarity between adjacent turns near each candidate boundary in $\mathcal{B}_1$. Boundaries with similarity below threshold $\tau$ form set $\mathcal{B}_2$, which helps determine the final topic boundaries $\mathcal{B}$.

## 3.2 LIGHT2: TOPIC-AWARE SHORT-TERM MEMORY

After obtaining individual topic segments, forming an index structure of $\{$topic, message turns$\}$, where message turns $= \{user_i, model_i\}$. These are first placed into the STM buffer. When the token count in the buffer reaches a preset threshold, we invoke LLM $f_{\text{sum}}$ to generate concise summaries of every structure. The final index structure stored in LTM is $\{$topic, $\{sum_i, user_i, model_i\}\}$.

$$\text{sum}_i = f_{\text{sum}}(S_i), \quad S_i \subseteq \{\text{user}_i, \text{model}_i\}, \ S_i \neq \varnothing,$$

$$\text{Entry}_i = \{\text{topic}, \mathbf{e}_i := \text{embedding}(\text{sum}_i), \text{user}_i, \text{model}_i\},$$

where $\text{Entry}_i$ denotes the memory entry to be stored in LTM. Compared with inputting at the granularity of a single turn or session, directly feeding multiple sessions can reduce subsequent API calls but often introduces inaccurate memory entries due to excessive topic mixing, leading to performance degradation. In contrast, topic-constrained input granularity minimizes API calls to the greatest extent while preserving summarization accuracy and maintaining stable system performance.

## 3.3 LIGHT3: LONG-TERM MEMORY WITH SLEEP-TIME UPDATE

**Soft Updating at Test Time.** At test time, when memory entries arrive, LightMem directly inserts them into LTM with soft updates, thereby decoupling the update process from online inference. Due to real-time updates being converted to direct insertions, interaction latency is significantly reduced. After all entries are inserted or when an update trigger arrives, we compute an update queue for every entry in LTM.

$$\mathcal{Q}(e_i) = \text{Top}_k \left\{ (e_j, \text{sim}(v_i, v_j)) \ | \ t_j \geq t_i, j \neq i \right\}_{:n},$$

where $e_i$ denotes the $i$-th memory entry with embedding $v_i$ and timestamp $t_i$, $\text{sim}(\cdot, \cdot)$ is the similarity function, and $\text{Top}_k\{\cdot\}_{:n}$ indicates selecting the top-$k$ most similar candidates, with the update queue $Q(e_i)$ length fixed at $n$. Consistent with existing work, we select the top-$k$ existing memory entries with the highest semantic similarity as potential update sources. On this basis, we further impose the constraint that only entries with later timestamps are allowed to update earlier ones ($t_j \geq t_i$), which is consistent with realistic temporal dynamics. Here, $\mathcal{Q}(e_i)$ denotes the queue of other entries that may update $e_i$. Since this process involves only similarity retrieval, it is fast and lightweight, and can be executed offline in parallel with online inference.

**Offline Parallel Update.** LightMem does not simply transfer online update latency to offline phases, it substantially reduces the overall update latency. The online update mechanism in existing memory frameworks enforces sequential updates, leading to a total latency that accumulates with each update. As shown in Figure 2, in LightMem, each memory entry maintains a global update queue, with each queue corresponding to a distinct $f_{\text{update}}$ operation. Since the update targets are independent across queues, updates can be executed in parallel, thereby greatly reducing the total latency.

## 4 COMPLEXITY ANALYSIS ABOUT LIGHTMEM

| Method | Summary Tokens | Update Tokens | API Calls | Runtime |
|---|---|---|---|---|
| Baselines | $N(L_{\text{sum-in}} + T + L_{\text{sum-out}})$ | $NM_1R_1(L_{\text{up-in}} + L_{\text{up-out}})$ | $N$ | $O(N)$ |
| LightMem | $\frac{Nr^xT}{th}(L_{\text{sum-in}} + th + L_{\text{sum-out}})$ | $\frac{Nr^xT}{th}M_2R_2(L_{\text{up-in}} + L_{\text{up-out}})$ | $\frac{Nr^xT}{th}$ | $O\left(\frac{Nr^xT}{th}\right)$ |

Table 1: Complexity comparison between LightMem and other memory systems. The specific definitions of each symbol are provided in the Appendix A.2.

As shown in Table 4, we consider a dialogue with $N$ turns, each containing on average $T$ tokens. In conventional memory systems, each turn triggers a summarization call, consuming $L_{\text{sum-in}} + T + L_{\text{sum-out}}$ tokens and totaling $N(L_{\text{sum-in}} + T + L_{\text{sum-out}})$ tokens with $N$ API calls. Each summarization

produces $M_1$ memory entries, a fraction $R_1$ of which retrieve at least one relevant neighbor and trigger an update, resulting in an update-token cost of $NM_1R_1(L_{\text{up-in}} + L_{\text{up-out}})$.

In **LightMem**, each turn is first passed through iterative pre-compression submodule, retaining only $r^xT$ tokens after $x$ iterations, and appended to a short-term memory (STM) buffer of capacity $th$. Summarization is triggered only when the buffer reaches capacity, yielding $\frac{Nr^xT}{th}$ summarization calls, each consuming $L_{\text{sum-in}} + th + L_{\text{sum-out}}$ tokens. Each summarization produces $M_2$ memory entries, but stricter retrieval constraints, including semantic similarity and timestamp filtering, reduce the fraction $R_2$ that trigger updates. Hence, the update phase involves $\frac{Nr^xT}{th}M_2R_2$ calls, with a total token cost of $\frac{Nr^xT}{th}M_2R_2(L_{\text{up-in}} + L_{\text{up-out}})$.

Overall, **LightMem** requires only $\frac{Nr^xT}{th}$ API calls for both summarization operations, substantially reducing token usage and call frequency compared to other systems. Correspondingly, the runtime complexity of other memory systems is $O(N)$, while LightMem achieves a reduced runtime of $O\left(\frac{Nr^xT}{th}\right)$, reflecting the efficiency gain from compressed summarization and selective updates.

## 5 EXPERIMENTS

### 5.1 EXPERIMENTAL SETUP

**Experimental Details.** (1) Our experiments adopt a realistic *Incremental Dialogue Turn Feeding* setting, where the entire dialogue history is fed and processed **at the turn level, one turn at a time**. This reflects practical scenarios where interactions between user and model is incrementally formed turn by turn. (Hu et al., 2025). (2) For considerations of both efficiency and effectiveness, we employ LLMLingua-2 as our pre-compressor throughout all subsequent experiments. (3) The attention scores for topic segmentation are also obtained using LLMLingua-2, the size of the sensory memory buffer is 512 tokens. All specific models used in this paper, can be found in Table 5.

**Datasets & Baseline Methods.** We use two well-known datasets, LONGMEMEVAL (Wu et al., 2025) (specifically the LongMemEval-S split) and LOCOMO (Maharana et al., 2024) to evaluate memory ability. We compare **LightMem** against several representative baselines of conversational memory modeling. ① *Full Text*, ② *Naive RAG*, ③ *LangMem* (LangChain, 2025), ④ *A-MEM* (Xu et al., 2025), ⑤ *MemoryOS* (Kang et al., 2025), ⑥ *Mem0* (Chhikara et al., 2025). In addition, all methods use GPT-4o-mini, Qwen3-30B-A3B-Instruct-2507 and GLM-4.6 as the LLM backbones. Details on dataset, baselines, and experimental settings are provided in the Appendix D.

**Metrics.** We evaluate these methods using both effectiveness and efficiency metrics. For effectiveness, we report **Accuracy (ACC)**, defined as the proportion of correctly answered questions. The evaluation is conducted with *GPT-4o-mini* as an LLM judge, guided by a detailed evaluation prompt (see Appendix E.1). For efficiency, we focus on tracking the computational costs of the LLM invocations in memory bank construction stage (see Section 2.1), all averaged across the entire dataset, as it is the one tied to the design and implementation differences of memory systems. The retrieval and usage stage is not our focus, because for fair comparison, The $f_{\text{retrieve}}()$, $f_{\text{chat}}()$ and number of retrieved entries are same among all methods. As a result, their costs exhibit only minor differences, and this stage is largely orthogonal to the design of memory systems, as shown in the table. Within the memory bank construction stage, only the two sub-processes **Summary** and **Update** involve the use of LLMs, $f_{\text{sum/extract}}()$ and $f_{\text{update}}()$. So for both processes, we report the token consumption from LLM calls, including input tokens, output tokens, and total token usage (in thousands). Additionally, we track **API Calls** counting the total number of LLM invocations, and **Runtime** recording the overall execution time for memory bank construction stage.

### 5.2 MAIN RESULTS

As shown in Table 2 and Table 3, **LightMem** demonstrates superior effectiveness and efficiency on both datasets across both GPT and Qwen backbones. For a fair comparison, all efficiency metrics for LightMem in the following analysis refer to the **combined online and offline** costs.

**LongMemEval.** On the LongMemEval benchmark, LightMem consistently outperforms the strongest baseline, A-Mem, in the ACC metric, improving accuracy by 2.09%–6.40% with GPT and

Table 2: Effectiveness and efficiency comparison on LONGMEMEVAL-S. The token usage is in thousands. – indicates no value for the metric. **Bold** denotes the best result, underline the second-best. $r$ denotes the compression rate. $th$ denotes the capacity threshold of the STM buffer, measured in tokens. Each pair of $r$ and $th$ corresponds to two rows: one for online soft update and one for offline update. OP-update denotes the offline parallel update process of **LightMem**.

| Method | ACC (%) | Summary Tokens (k) | | Update Tokens (k) | | Total (k) | Calls | Runtime (s) |
| --- | --- | --- | --- | --- | --- | --- | --- | --- |
| | | In | Out | In | Out | | | |
| 🟢 **GPT-4o-mini** | | | | | | | | |
| FullText | 56.80 | – | – | – | – | 105.07 | – | – |
| NaiveRAG | 61.00 | – | – | – | – | – | – | 867.38 |
| LangMem | 37.20 | – | – | 982.68 | 119.48 | 1,102.16 | 520.62 | 2,293.70 |
| A-MEM | 62.60 | 214.66 | 42.82 | 1,157.52 | 190.81 | 1,605.81 | 986.55 | 5,132.06 |
| MemoryOS | 44.80 | 2,302.35 | 304.18 | 350.02 | 35.19 | 2,991.75 | 2,938.41 | 8,030.04 |
| Mem0 | 53.61 | 424.13 | 17.76 | 560.17 | 150.56 | 1,152.62 | 811.57 | 4,248.49 |
| LightMem | | | | | | | | |
| $r$=0.5, $th$=256 | 64.29 | 20.80 | 10.01 | – | – | 30.81 | 25.67 | 302.69 |
| (OP-update) | 64.69 | – | – | **44.46** | **2.56** | 47.02 | 70.23 | 342.63 |
| $r$=0.6, $th$=256 | 67.78 | 24.58 | 10.53 | – | – | 35.11 | 30.47 | 329.61 |
| (OP-update) | 65.39 | – | – | 53.98 | 3.18 | 57.16 | 85.07 | 411.56 |
| $r$=0.7, $th$=512 | **68.64** | 18.88 | 9.37 | – | – | **28.25** | **18.43** | **283.76** |
| (OP-update) | 67.07 | – | – | 79.38 | 4.06 | 83.44 | 125.47 | 496.03 |
| 🔷 **Qwen3-30B-A3B-Instruct-2507** | | | | | | | | |
| FullText | 54.80 | – | – | – | – | 105.07 | – | – |
| NaiveRAG | 60.80 | – | – | – | – | – | – | 659.09 |
| LangMem | 50.80 | – | – | 1,311.96 | 118.06 | 1,430.02 | 495.12 | 3,237.16 |
| A-MEM | 65.20 | 219.21 | 66.98 | 1,260.54 | 318.20 | 1,864.93 | 989.30 | 5,367.51 |
| MemoryOS | 49.60 | 2,101.54 | 510.88 | 305.12 | 27.43 | 2,944.97 | 2,922.28 | 8,721.78 |
| Mem0 | 39.51 | 424.20 | **15.34** | 411.50 | 111.35 | 1001.90 | 722.76 | 2,239.94 |
| LightMem | | | | | | | | |
| $r$=0.4, $th$=768 | 61.95 | **9.01** | 16.14 | – | – | **25.15** | 16.54 | 357.13 |
| (OP-update) | 62.34 | – | – | 111.13 | 7.88 | 119.01 | 176.02 | 1036.47 |
| $r$=0.6, $th$=768 | **70.20** | 13.19 | 19.21 | – | – | 32.40 | 19.97 | 417.13 |
| (OP-update) | 65.14 | – | – | **97.11** | **5.92** | 103.03 | 152.93 | 1023.56 |
| $r$=0.8, $th$=1024 | 68.69 | 14.82 | 18.49 | – | – | 33.31 | **9.43** | **355.71** |
| (OP-update) | 67.34 | – | – | 106.91 | 6.20 | 113.11 | 168.37 | 1026.90 |
| 🅩 **GLM-4.6** | | | | | | | | |
| FullText | 36.71 | – | – | – | – | 103.38 | – | – |
| NaiveRAG | **73.20** | – | – | – | – | – | – | 53,725.15 |
| LangMem | 49.20 | – | – | 3,052.42 | 7.03 | 3,059.45 | 314.61 | 5,577.91 |
| A-MEM | 70.60 | 444.95 | 63.40 | 1,992.04 | 403.39 | 2,903.78 | 450.40 | 8,068.80 |
| LightMem | | | | | | | | |
| $r$=0.5, $th$=256 | 73.00 | 16.55 | 14.06 | – | – | 30.61 | 10.78 | **1,014.37** |
| $r$=0.6, $th$=256 | **73.20** | **16.55** | 13.99 | – | – | 30.54 | **10.78** | 1,077.69 |
| $r$=0.7, $th$=512 | 72.80 | **16.55** | **13.91** | – | – | **30.46** | 10.78 | 1,038.19 |

up to 7.67% with Qwen. In terms of efficiency, for GPT, LightMem reduces total token consumption by 10×–38× and API calls by 3.6×–30×; for Qwen, it reduces total tokens by 6.9×–21.8× and API calls by 3.3×–17.1×. Regarding runtime, LightMem achieves 2.9×–12.4× for GPT and 1.6×–6.3× for Qwen speedup over other memory baselines.

If considering only online test-time cost, LightMem shows an even larger efficiency advantage. For GPT, LightMem reduces total token consumption by 31.4×–105.9× and API calls by 17.1×–159.4×; for Qwen, it reduces total tokens by 30.1×–117.1× and API calls by 24.8×–309.9×.

**LoCoMo.** On the LoCoMo dataset, LightMem also demonstrates superior performance over other memory baselines. For the GPT backbone, it improves ACC by 6.10%–18.12%, achieves a 2.87×–

Table 3: Effectiveness and efficiency comparison on LOCOMO. Due to space limitations and for ease of comparison, we merge the results before and after LightMem's offline update into a single row. The ACC reported corresponds to the performance after the offline update.

| Method | ACC (%) | Summary Tokens (k) | | Update Tokens (k) | | Total (k) | Calls | Runtime (s) |
|---|---|---|---|---|---|---|---|---|
| | | In | Out | In | Out | | | |
| 🔮 GPT-4o-mini | | | | | | | | |
| FullText | 71.83 | – | – | – | – | – | – | – |
| NaiveRAG | 63.64 | – | – | – | – | – | – | – |
| LangMem | 57.20 | – | – | 898.27 | 111.95 | 1010.22 | 920.62 | 2229.37 |
| A-MEM | 64.16 | 182.74 | 49.29 | 729.89 | 187.52 | 1149.43 | 1175.47 | 6060.73 |
| MemoryOS(locomo)[1] | 58.25 | 110.98 | 33.40 | 78.08 | 64.54 | 287.00 | 553.45 | 2422.05 |
| MemoryOS(regular) | 54.87 | 226.86 | 46.61 | 177.66 | 75.34 | 526.48 | 1016.06 | 3332.59 |
| Mem0 | 61.69 | 851.32 | 20.53 | 632.12 | 189.42 | 1693.39 | 1602.20 | 4432.87 |
| LightMem(0.7,512) | 71.95 | 73.19 | 20.13 | 6.05 | 0.40 | 99.76 | 41.65 | 848.49 |
| LightMem(0.7,768) | 70.26 | **57.54** | 18.92 | **3.79** | **0.23** | **80.48** | **29.55** | **737.80** |
| LightMem(0.8,768) | **72.99** | 62.82 | **17.95** | 4.14 | 0.28 | 85.19 | 29.83 | 815.32 |
| 🔮 Qwen3-30B-A3B-Instruct-2507 | | | | | | | | |
| FullText | **74.87** | – | – | – | – | – | – | – |
| NaiveRAG | 66.95 | – | – | – | – | – | – | – |
| LangMem | 60.53 | – | – | 1004.35 | 138.02 | 1142.37 | 1005.37 | 2268.57 |
| A-MEM | 56.10 | 158.29 | 60.85 | 924.19 | 483.51 | 1626.80 | 1175.40 | 5543.90 |
| MemoryOS(locomo) | 61.04 | 122.21 | 53.12 | 104.43 | 81.75 | 361.51 | 414.70 | 1269.70 |
| MemoryOS(regular) | 51.30 | 228.85 | 51.60 | 242.27 | 143.63 | 666.35 | 1004.60 | 1982.20 |
| Mem0 | 43.31 | 827.09 | **18.64** | 763.88 | 189.80 | 1799.40 | 1614.50 | 4540.70 |
| LightMem(0.6,768) | 71.36 | **56.68** | 34.14 | **8.31** | **0.74** | **99.87** | **29.10** | **815.70** |
| LightMem(0.8,1024) | 72.60 | 61.38 | 36.33 | 9.86 | 0.88 | 108.45 | 32.00 | 1079.40 |

20.92× improvement in total token efficiency, reduces API calls by 13.29×–39.78×, and accelerates runtime by 2.63×–8.21×. On the Qwen backbone, LightMem maintains its advantage in both effectiveness and efficiency, with 4.41%–29.29% higher ACC, 3.33×–18.02× reduction in total token consumption, 12.96×–55.48× fewer API calls, and 1.18×–5.57× faster runtime.

**LightMem achieves superior performance on nearly all metrics and both LLM backbones, while demonstrating robust performance and efficiency on both LongMemEval and LoCoMo, highlighting its generalizability across different models and scenarios.**

## 5.3 ANALYSIS OF PRE-COMPRESSING SUBMODULE

**Performance and Overhead.** LightMem uses an additional model (Pan et al., 2024b; Xia et al., 2025) for pre-compression. We evaluate its performance by randomly sampling 1/5 of LONG-MEMEVAL and compressing it at ratios shown in Figure 3(a), then prompting LLMs for in-context QA. When compression ratio $r$ ranges from 50%–80%, compressed and uncompressed performance are comparable, demonstrating LLMs can effectively understand compressed content and validating LightMem's approach. The submodule is highly efficient, consuming under 2GB of GPU memory with negligible impact on overall runtime.

**Impact of $r$ on Performance.** As shown in Tables 8 and 9, The optimal $r$ for ACC is dependent on the STM buffer threshold $th$. For smaller thresholds ($th \in \{0, 256\}$), an $r$ of 0.6 achieves the highest ACC. In contrast, for larger thresholds ($th \in \{512, 1024\}$), a higher retention rate of $r = 0.7$ performs best. This suggests greater buffer capacity enables effective use of richer, less-compressed information, leveraging LLMs' advanced long-context processing to mitigate the "lost in the middle" phenomenon. On average, the optimal $r$ for ACC is 0.6, reflecting a trade-off between information compression rate and the quantity of information in the STM buffer. In terms of efficiency, a lower $r$

---

[1]MemoryOS(locomo) is the LoCoMo reproduction script in the MemoryOS library, simplifying the standard version, shown as MemoryOS(regular).

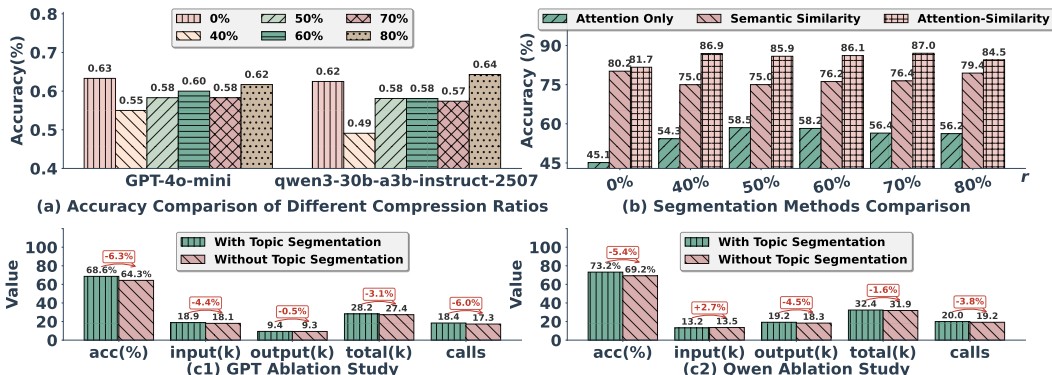

Figure 3: Analysis and Ablation Study of Key Modules. Fig.(a) depicts the QA accuracy when using prompts compressed at different ratios ($r$) as in-contexts to query the LLM directly. Fig.(b) compares the accuracy of different topic segmentation methods under these varying compression ratios. Fig.(c1) and Fig.(c2) present the ablation study for the topic segmentation module, evaluating its impact on both performance and efficiency for the GPT and Qwen models.

generally leads to higher efficiency, as it triggers the buffer threshold less frequently under the same $th$, resulting in fewer API calls and lower token consumption.

## 5.4 ANALYSIS OF TOPIC SEGMENTATION SUBMODULE

**Segmentation Accuracy.** To validate the accuracy of our proposed hybrid topic segmentation method, we compare it with segmentation using only a single granularity: attention-only-based and similarity-only-based segmentation. Since the construction process of the LONGMEMEVAL indicates that different sessions naturally serve as topic boundaries, we directly use them as ground-truth labels. The final accuracy is calculated as the number of correctly identified segmentation points divided by the total number of labels. The results in Figure 3(b) validate the effectiveness of our method: it achieves higher accuracy than both individual segmentation methods across all compression ratios, with an absolute accuracy exceeding 80%.

**Ablation Study.** As shown in Figure 3(c), removing the topic segmentation submodule slightly improves efficiency but significantly harms accuracy, causing a 6.3% drop for GPT and 5.4% for Qwen. This indicates that the submodule effectively enables models to perceive semantic units in the input, facilitating subsequent memory unit generation.

## 5.5 ANALYSIS OF THE STM THRESHOLD'S IMPACT

As illustrated in the Figure 4, the STM buffer threshold ($th$) has a distinct but significant impact on both efficiency and performance metrics. A consistent trend is: as $th$ increases, there is a marked improvement in efficiency. In contrast, the effect on QA accuracy is non-monotonic. The optimal threshold for accuracy varies depending on the model and the compression ratio ($r$), indicating that a larger buffer does not always yield better performance. This highlights a crucial trade-off: while a larger STM threshold is consistently better for reducing computational cost, the ideal setting for maximizing task accuracy requires careful tuning.

## 5.6 ANALYSIS OF SLEEP-TIME UPDATE

**Why Soft Updates Work.** A primary challenge in designing memory systems is handling updates. While powerful, LLMs can be unreliable when tasked with complex real-time update operations. For instance, when presented with two related but not contradictory pieces of information, an LLM might incorrectly interpret them as a conflict and delete the older memory entry, leading to irreversible information loss. Instead, the optimal operations might be to merge the information or

simply add the new entry. In contrast, **LightMem** performs only incremental additions through soft updates during test time, which preserves global information and complete semantics.

---

**Case Study: Memory Update Mechanism Comparison**

**History1:** {'Monday, 2 PM': User is planning a trip to Tokyo.}
**History2:** {'Monday, 4 PM': User asks about trains to Kyoto.}
**Hard Update:** Overwrites memory          **LightMem Soft Update:** Appends info
-> "User plans Kyoto trip"                   -> "Tokyo trip + Kyoto inquiry"
⚠ Tokyo context lost                         ✅ Full context preserved

---

## 6  RELATED WORK

**Hard Prompt Compression for LLMs.** Hard prompt compression improves LLM efficiency by removing redundant content from prompts (Li et al., 2025c). Methods recently have evolved from using smaller language models (Jiang et al., 2023; Li et al., 2023; Chuang et al., 2024) to query-aware approaches that preserve task-relevant information (Weston & Sukhbaatar, 2023; Creswell et al., 2023; Jiang et al., 2024). Additionally, lightweight bidirectional encoders have demonstrated strong effectiveness and efficiency (Pan et al., 2024a; Liskavets et al., 2025).

**Chunking Strategies in RAG Systems.** Retrieval-Augmented Generation (RAG) systems rely on chunking extrernal documents into smaller units for retrieval (Lewis et al., 2020; Gao et al., 2023). Existing chunking strategies include rule-based methods creating fixed-size segments (Lewis et al., 2020; Sarthi et al., 2024; Edge et al., 2024; Gutierrez et al., 2024), semantic-based methods grouping content by topic (Qu et al., 2025), and LLM-driven methods leveraging model knowledge for splitting (Pan et al., 2025; Duarte et al., 2024; Zhao et al., 2024; Liu et al., 2025b). However, all of these chunking strategies for RAG systems are tailored to static scenarios, not applicable to dynamic and open-ended environments.

**Memory Systems for LLM Agents.** Memory systems help LLM agents move beyond stateless interactions to support flexible reasoning and adaptation in complex and changing environments (Liu et al., 2025a; Mei et al., 2025). The earliest and most straightforward approaches store experiences as linear or sequential streams, sometimes enhanced with hierarchical structures (Liang et al., 2023; Park et al., 2023; Packer et al., 2023; Zhong et al., 2024; Salama et al., 2025; Fang et al., 2025). A more structured class of methods represents memories as nodes and their relationships as edges, using trees, graphs, or temporal knowledge structures to support retrieval and update (Rezazadeh et al., 2025; Chhikara et al., 2025; Rasmussen et al., 2025; Xu et al., 2025; Zhang et al., 2025). The latest trend integrates various types of memory, allowing them to interact and synergistically improve overall performance (Kang et al., 2025; Li et al., 2025b; Wang & Chen, 2025; Nan et al., 2025). Overall, existing memory systems for LLM agents have become increasingly complex and capable, leveraging hierarchical, structured, and multi-type memories. However, most focus on maximizing effectiveness, with limited consideration of efficiency. While some recent works (Guo et al., 2024; Zhao et al., 2025; Dong et al., 2025) share a similar motivation with our work, they focus on lightweight adaptations of GraphRAG where the corpus is predefined and static.

## 7  CONCLUSION

In this work, we introduced LightMem, a lightweight and efficient memory framework designed to address the significant overhead of memory systems for LLM agents. Inspired by the multi-stage Atkinson-Shiffrin human memory model, LightMem's architecture effectively filters, organizes, and consolidates information. Our empirical evaluation demonstrates that this approach maintains strong task performance while sharply reducing computational costs. In the near future, we plan to accelerate LightMem's update phase via offline pre-computed KV caches, reducing runtime overhead. We aim to integrate a lightweight knowledge graph memory for explicit multi-hop reasoning and structured retrieval. A multimodal memory extension will enable adaptation to visual, auditory, and textual inputs in embodied and real-world scenarios.

ETHICS STATEMENT

LightMem enhances LLM agents by creating an external memory of user interactions. While this improves agent coherence, it introduces critical ethical challenges. Storing dialogue histories poses inherent risks to user privacy, as conversations may contain sensitive data. The memory can also absorb and perpetuate biases or misinformation from user input, potentially leading to bad agent behavior. Therefore, any deployment of this technology must prioritize robust safeguards. We strongly advocate for strict privacy protocols, such as data anonymization and user consent, as well as mechanisms to mitigate the effects of biased or false memories. Responsible development is essential to ensure these memory-augmented systems are used in a safe and trustworthy manner.

REPRODUCIBILITY STATEMENT

To ensure the reproducibility of this work, we introduce the detailed implementations for LightMem are provided in in Section 3, Appendix C. Additionally, we plan to release our source code in the future to further support reproducibility. These measures are intended to facilitate the verification and replication of our results by other researchers in the field.

ACKNOWLEDGMENTS

We would like to express our sincere gratitude to the anonymous reviewers for their thoughtful and constructive feedback. This work was supported by the National Natural Science Foundation of China (No. 62576307, No. NSFCU23B2055, No. NSFCU19B2027), the Fundamental Research Funds for the Central Universities (226-2023-00138), the 2025 Zhejiang Provincial Center for Disease Control and Prevention Science and Technology Talent Incubation Project (No. 2025-A-04), the 2025 Zhejiang Health Informatics Association Scientific Research Program (Key Project, No. 2025XHZN-Z01), titled "Research on Monitoring and Early Warning Methods of AI Large Model and Infectious Disease Epidemic Data Fusion", undertaken by the Zhejiang Provincial Center for Disease Control and Prevention, Yongjiang Talent Introduction Programme (2021A-156-G), and Information Technology Center and State Key Lab of CAD&CG, Zhejiang University.

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

# A BACKGROUND DETAILS

## A.1 BACKGROUND ABOUT CURRENT MEMORY SYSTEMS

We describe both the mainstream memory architectures and the **LightMem** pipeline in terms of two major stages. The first is the memory bank construction stage, which can be further decomposed into the three sub-stages (I), (II), and (III) described in the Section 2.1. The second major stage concerns the usage of the memory system, which consists of retrieval and question answering (QA).

**Memory Bank Construction**    As shown in Table 4, we detail the workflows of the three sub-stages (I), (II), and (III) for naive RAG, prevailing memory systems, and our LightMem. It can be observed that baseline memory systems typically perform their update stage during user–model interaction, which introduces substantial test-time latency. In contrast, LightMem decouples this update process from online interaction, thereby significantly reducing test-time latency. All models involved in these processes are listed in Table 5. As shown, LightMem introduces only one additional model, LLMlingua-2,beyond those used by baseline methods. This model follows a lightweight BERT architecture and requires less than 2GB of GPU memory during inference, rendering its overhead negligible. Moreover, for fairness, the latency introduced by this component is fully accounted for in our reported Runtime metric.

Table 4: The mainstream memory architectures and the LightMem pipeline of memory bank construction stage. Black-font processes denote those executed during online test-time interactions, whereas red-font processes denote those executed offline.

| Method | (I) Segment | (II) Summary/Extrct | (III) Update |
|---|---|---|---|
| NaiveRAG | Raw dialog $\rightarrow f_{\text{seg}}()$ $\rightarrow \{\text{seg}_i\}$ | $\rightarrow f_{\text{index}}() \rightarrow \{\text{emb}_i\}$ | \ |
| Other Memory Systems | Raw dialog $\rightarrow f_{\text{seg}}()$ $\rightarrow \{\text{seg}_i\}$ | $\rightarrow f_{\text{sum/extract}}() \rightarrow \{\text{memory entry}_i\}$ $\rightarrow f_{\text{index}}() \rightarrow \{\text{emb}_i\}$ | $\rightarrow f_{\text{retrieve}}() \rightarrow \{\text{related entry}_i\}$ $\rightarrow f_{\text{update}}()$ $\rightarrow \{\text{add, delete, update, merge...}\}$ |
| LightMem | Raw dialog $\rightarrow f_{\text{seg}}()$ $\rightarrow \{\text{seg}_i\}$ $\rightarrow f_{\text{pre\_compress}}()$ $\rightarrow \{\text{comp\_seg}_i\}$ $\rightarrow$ sensory buffer full $\rightarrow$ $f_{\text{topic}}() \rightarrow$ $\{\text{topic-wise comp\_seg}_i\}$ | $\rightarrow$ STM buffer full $\rightarrow f_{\text{sum/extract}}()$ $\rightarrow \{\text{topic}_i, \{\text{memory entry}_j\}\}$ $\rightarrow f_{\text{index}}() \rightarrow \{\text{topic}_i, \{\text{emb}_j\}\}$ | Offline update trigger $\{\text{every entry}_i\} \rightarrow f_{\text{retrieve}}()$ $\rightarrow \{\text{related entry}_j\} \rightarrow \{\text{update queue}\}$ All update queues established $\rightarrow$ parallel $f_{\text{update}}()$ $\rightarrow \{\text{add, delete, update, merge...}\}$ |

| Function | Model / Strategy | Implementation in This Paper |
|---|---|---|
| $f_{\text{seg}}()$ | Segmentation strategy | Turn-level granularity input |
| $f_{\text{index}}()$ | Embedding model | all-MiniLM-L6-v2 |
| $f_{\text{sum/extract}}()$ | System backbone model | GPT-4o-mini; Qwen3-30B-A3B-Instruct-2507 |
| $f_{\text{retrieve}}()$ | Retrieval strategy | Cosine similarity vector retrieval |
| $f_{\text{update}}()$ | System backbone model | GPT-4o-mini; Qwen3-30B-A3B-Instruct-2507 |
| $f_{\text{pre\_compress}}()$ | Token compression model | LLMlingua-2 |
| $f_{\text{topic}}()$ | Topic segmentation model | LLMlingua-2 |
| $f_{\text{chat}}()$ | Chat model | GPT-4o-mini; Qwen3-30B-A3B-Instruct-2507 |

Table 5: Mapping between functions, their roles, and the concrete models used in this paper. Black-font entries denote models shared by both LightMem and baseline methods, whereas red-font entries denote models unique to LightMem.

**Retrieval and Usage**    After the memory bank construction stage, we obtain an up-to-date memory bank. When a new user query arrives, the memory system use $f_{\text{retrieve}}()$ to retrieve relevant entries from this repository, appends them to the query, and then prompts the chat model $f_{\text{chat}}()$ to produce a response.

## A.2 Notation and Complexity Details

Table 6: Notation used in complexity analysis (§Section 4).

| Symbol | Definition |
| --- | --- |
| $N$ | Total number of turns in a dialogue history. |
| $T$ | Average number of tokens per turn. |
| $r$ | Token compression rate (as defined in the main paper). After one compression step, only a fraction $r$ of tokens is retained. |
| $x$ | Number of compression iterations. In LightMem, the *pre-compress* module may be invoked multiple times for the same message to remove redundancy until the message is sufficiently compact. This occurs frequently in datasets such as **LongMemEval**. All time costs are included in runtime metrics. |
| $th$ | Capacity of the Short-Term Memory (STM) buffer, as defined in the paper. |
| $L_{\text{sum-in}}$ / $L_{\text{sum-out}}$ | Number of tokens in the **input prompt template** and **output** of a single backbone LLM call for *summarization*. These are similar across memory frameworks. |
| $M_1$ / $M_2$ | Number of memory entries produced from a single summarization operation under Other Memory Systems ($M_1$) and LightMem ($M_2$). |
| $L_{\text{up-in}}$ / $L_{\text{up-out}}$ | Number of tokens in the **input prompt template** and **output** of a single backbone LLM call for *memory update*. Similar across frameworks. |
| $R_1$ / $R_2$ | Proportion of summary entries that successfully retrieve at least one relevant memory entry (triggering an update) for Other Memory Systems ($R_1$) and LightMem ($R_2$). Some entries do not retrieve any relevant counterparts and thus do not trigger updates. |

## B Usage of LLMs

Throughout the preparation of this manuscript, we used LLMs to assist with improving grammar, clarity, and wording in parts of this work. The use of LLMs was limited to language refinement, with all ideas, analyses, and conclusions solely developed by the authors.

## C Methodology Details

### C.1 Topic Segmentation

In this part, we present the construction of the attention matrix, the underlying rationale for topic segmentation, and representative illustrative cases.

We extract only the user sentences from multi-turn dialogues, as they are generally more concise and the assistant's responses necessarily remain consistent with the user's theme within the same turn. Moreover, since the maximum input length of the LLMLingua-2 Pan et al. (2024b) model is 512 tokens, the assistant's often lengthy sentences cannot be effectively accommodated. Therefore, we sequentially store the user sentences into a buffer and segment them, ensuring that as many sentences as possible are preserved while staying within the token limit. As a practical trick, if a sentence becomes empty after compression, we retain its original uncompressed version; if the token length of a sentence still exceeds the maximum limit, we continue to compress it using the LLMLingua-2 model at a 0.5 compression rate until the token length falls below the threshold. To reduce the effect of attention sinks, we mask out the contributions of the first and last three tokens in each sequence and subsequently normalize the remaining attention values. Attention is derived from the higher layers of LLMLingua-2 (layers 8, 9, 10, and 11). For any two sentences, we first compute token-level pairwise attention and average across tokens to obtain the overall attention of one sentence to the target sentence; we then average across the selected layers to obtain a more robust inter-sentence attention score. For each current sentence, the attention scores directed toward all

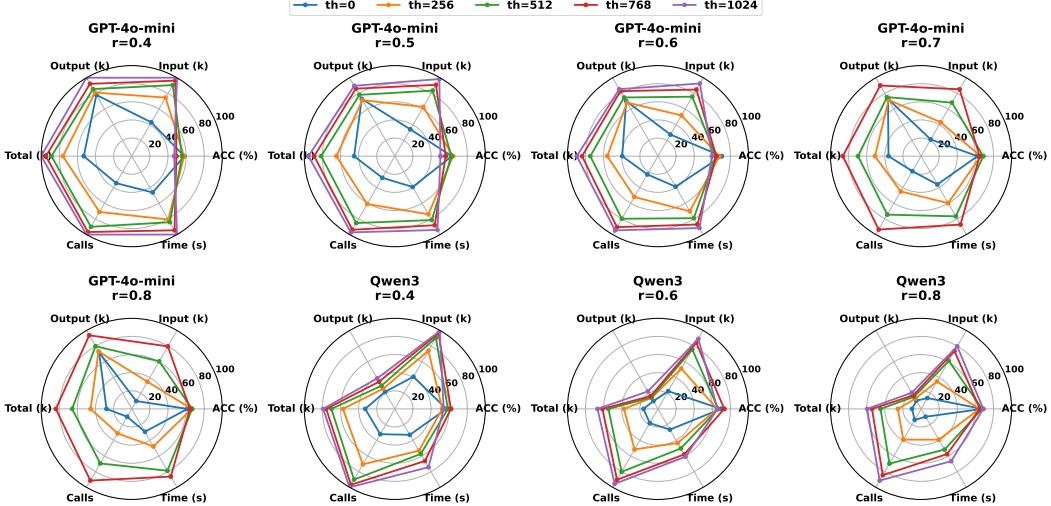

Figure 4: Impact of the STM buffer threshold ($th$) on performance and efficiency across different compression ratios ($r$). Each radar chart represents a specific configuration of a model (GPT-4o-mini or Qwen3) and a fixed compression ratio. The axes measure six key metrics: Accuracy (ACC), token consumption (Input, Output, Total), API Calls, and Runtime. To facilitate comparison, all values are normalized for visualization on the chart.

preceding sentences are normalized within the sentence, yielding the final attention matrix. Residual fragments that remain after segmentation are carried over to the beginning of the next buffer for further processing, and this procedure continues iteratively until the dialogue ends.

Based on the attention pattern, we focus on the sequence formed by each sentence's attention scores relative to its immediately preceding sentence, which directly reflects the continuity of local semantics. Therefore, we take the attention scores from the outermost layer of the attention map. When the attention score at a given position is higher than both its preceding and following positions, it is regarded as a local peak. If a sentence is identified as a peak, we set a segmentation point immediately before this sentence, making the peak sentence the beginning of a new segment. The rationale is that the peak sentence exhibits consistently low attention to all earlier sentences overall and reflects a clear transition from an old topic to a new one, indicating that the identified sentence marks the initiation of a new topic.

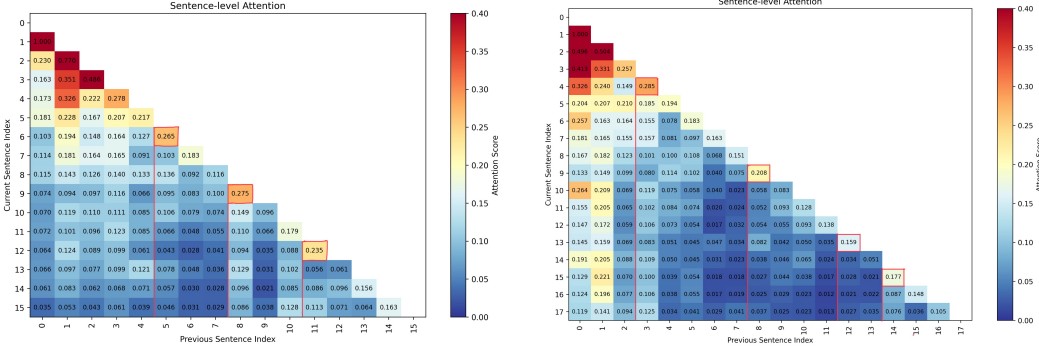

Figure 5: Example of Topic Segment Attention Matrix.

Figure 5 illustrates three representative examples of reliable segmentation under 50% compression rate. In the first attention map, local peaks in the adjacent-sentence attention sequence appear at positions 5, 8, and 11, where the actual segmentation boundaries lie between sentences 4–5 and 11–12.

In the second attention map, peaks occur at positions 3, 8, 12, and 14, and the actual boundaries are located between sentences 7–8, 11–12, and 13–14. Overall, our method achieves close alignment with the majority of true boundaries while providing finer-grained segmentation. These examples demonstrate that our segmentation approach enables both fine-grained and reliable detection of topic boundaries, thereby validating its effectiveness.

## C.2 CATEGORY-WISE ACCURACY

As summarized in Table 7, retrieval-augmented and memory-centric methods (e.g., *A-MEM*, *Mem0*, *MemoryOS*) generally outperform *Full Text* on categories that demand information integration or belief revision, such as *Temporal*, *Multi-Session*, and *Knowledge-Update*. In contrast, categories such as *Single-User* and *Single-Assistant*, lightweight retrieval like *Naive RAG* is often competitive and can be the most reliable option, while *Single-Preference* shows higher variance due to its smaller sample size.

Table 7: **Category-wise Accuracy**. Accuracy (%) by method across question types. Parentheses indicate category proportion and sample size. For GPT, LightMem is configured with parameters $r = 0.7$ and $th = 512$; for Qwen, LightMem is configured with $r = 0.4$ and $th = 768$.

| Method | Temporal ($n$=133) | Multi-Session ($n$=133) | Knowledge-Update ($n$=78) | Single-User ($n$=70) | Single-Assistant ($n$=56) | Single-Preference ($n$=30) |
|---|---|---|---|---|---|---|
| | | | GPT-4o-mini | | | |
| Full Text | 31.58 | 45.45 | 76.92 | 87.14 | 89.29 | 36.67 |
| Naive RAG | 39.85 | 48.48 | 67.95 | 90.00 | 98.21 | 53.33 |
| LangMem | 15.79 | 20.30 | 66.67 | 60.00 | 46.43 | 60.00 |
| A-MEM | 47.36 | 48.87 | 64.11 | 92.86 | 96.43 | 46.67 |
| MemoryOS | 32.33 | 31.06 | 48.72 | 80.00 | 64.29 | 30.00 |
| Mem0 | 40.15 | 46.21 | 70.12 | 81.43 | 41.07 | 60.00 |
| LightMem | 67.18 | 71.74 | 83.12 | 87.14 | 32.14 | 68.18 |
| | | | Qwen3-30B-A3B-Instruct-2507 | | | |
| Full Text | 33.08 | 35.61 | 76.92 | 82.86 | 87.50 | 50.00 |
| Naive RAG | 36.84 | 47.73 | 65.38 | 91.43 | 98.21 | 70.00 |
| LangMem | 37.60 | 38.35 | 67.95 | 78.57 | 42.86 | 70.00 |
| A-MEM | 51.88 | 51.12 | 76.93 | 90.00 | 96.43 | 40.00 |
| MemoryOS | 28.57 | 36.84 | 61.54 | 72.86 | 92.86 | 33.33 |
| Mem0 | 41.94 | 28.13 | 28.57 | 55.32 | 26.09 | 81.82 |
| LightMem | 54.20 | 51.91 | 66.67 | 80.00 | 31.25 | 80.00 |

## C.3 DETAILED PARAMETER ANALYSIS

As Table 9 shows, we report the numerical results of the effects of LightMem parameters (compression ratio $r$ and STM threshold $th$).

# D EXPERIMENT DETAILS

## D.1 DATASETS AND BASELINES

**Datasets** The LongMemEval dataset (Wu et al., 2025) is designed to benchmark long-term interactive memory in conversational agents. It comprises 500 evaluation questions built upon extended user-assistant dialogues. It has two different versions with different lengths: the LONGMEMEVAL-S setting contains approximately 115k tokens per problem, while the LONGMEMEVAL-M setting extends up to 1.5 million tokens across 500 sessions. In our work, we adopt the LONGMEMEVAL-S version due to its balance between dialogue length and computational feasibility. The questions are categorized into multiple types: information extraction, multi-session reasoning, knowledge updates, temporal reasoning, and abstention. Overall, the dataset is characterized by extremely long

Table 8: The impact of **LightMem** compression ratio $r$ and STM buffer threshold $th$ is reported here. Due to space limitations, we only present a subset of representative results of the online soft update results, with more results provided in the Figure 9.

| Model | $th$ | $r$ | ACC | Input (k) | Output (k) | Total (k) | Calls | Time |
|---|---|---|---|---|---|---|---|---|
| GPT | 256 | 0.5 | 64.29 | **20.80** | 10.01 | **30.81** | **25.67** | **302.69** |
| | 256 | 0.6 | **67.68** | 24.58 | 10.53 | 35.11 | 30.47 | 329.61 |
| | 256 | 0.7 | 65.68 | 27.66 | **9.97** | 37.63 | 34.26 | 403.59 |
| | 512 | 0.6 | 63.74 | **16.23** | 9.45 | **25.68** | **15.63** | **266.98** |
| | 512 | 0.7 | **68.64** | 18.88 | 9.37 | 28.25 | 18.43 | 283.76 |
| | 512 | 0.8 | 66.67 | 21.55 | **8.59** | 30.14 | 21.11 | 268.97 |
| | 1024 | 0.6 | 59.68 | **10.34** | 7.68 | **18.20** | **7.69** | **177.45** |
| | 1024 | 0.7 | **64.68** | 12.93 | 6.90 | 19.83 | 8.25 | 209.12 |
| | 1024 | 0.8 | 64.35 | 14.86 | **6.28** | 21.14 | 9.43 | 216.08 |
| Qwen | 512 | 0.4 | 58.57 | **11.03** | **17.00** | **28.03** | **10.11** | **421.74** |
| | 512 | 0.6 | 66.57 | 16.22 | 19.50 | 35.72 | 15.40 | 471.09 |
| | 512 | 0.8 | **67.37** | 21.35 | 19.36 | 40.71 | 20.98 | 461.02 |
| | 768 | 0.4 | 61.95 | **9.01** | **16.14** | **25.15** | **6.54** | **357.13** |
| | 768 | 0.6 | **73.20** | 13.19 | 19.21 | 32.40 | 9.97 | 417.13 |
| | 768 | 0.8 | 64.95 | 16.94 | 19.06 | 36.00 | 13.09 | 420.14 |
| | 1024 | 0.4 | 53.91 | **8.02** | **15.44** | **23.46** | **4.83** | **300.56** |
| | 1024 | 0.6 | 65.67 | 11.50 | 18.21 | 29.71 | 7.18 | 396.35 |
| | 1024 | 0.8 | **68.69** | 14.82 | 18.49 | 33.31 | 9.43 | 355.71 |

histories, wide temporal spans, and diverse question types, making it a comprehensive benchmark for evaluating conversational agents' memory capabilities. During the experiments, five samples from this dataset contained corrupted characters, which caused LightMem's compression model to fail to run properly. Consequently, LightMem directly discarded these five samples when processing the dataset. However, their accuracy results were uniformly treated as false. The indices of these five samples in the dataset are 74, 183, 278, 351, and 380.

The LoCoMo benchmark targets the evaluation of long-range conversational memory. It features extremely long dialogues, with each conversation spanning roughly 300 turns and around 9K tokens on average. The accompanying questions fall into four categories—Single-hop, Multi-hop, Temporal, and Open-domain—providing a comprehensive assessment of different dimensions of memory in LLMs.

**Baselines** We compare our approach against several representative baselines of conversational memory modeling. ① LANGMEM (LangChain, 2025): The Langchain's long-term memory module. ② A-MEM (Xu et al., 2025): Constructs a memory-centric knowledge graph, encoding each interaction as a structured memory note and linking these notes via LLM-driven reasoning. ③ MEM-ORYOS (Kang et al., 2025): Organizes conversational memory in an OS-inspired hierarchy, structuring interactions into short-term, mid-term, and long-term layers via paging and heat-based updating. ④ MEM0 (Chhikara et al., 2025): Extracts memories from dialogue turns through a combination of global summaries and recent context, maintaining them via LLM-guided operations.

## D.2 IMPLEMENTATION DETAILS

All the experiments are conducted on hardware equipped with 4×NVIDIA RTX 3090 GPUs, dual Intel Xeon Gold 6133 CPUs (40 cores, 80 threads), and 256 GB of RAM.

Table 9: The impact of LightMem's compression ratio ($r$) and STM buffer threshold ($th$).

| Model | $th$ | $r$ | ACC | Input (k) | Output (k) | Total (k) | Calls | Time |
|---|---|---|---|---|---|---|---|---|
| GPT-4o-mini | 0 | 0.4 | 58.04 | 27.70 | 8.90 | 36.60 | 39.91 | 500.69 |
| | 256 | 0.4 | 57.78 | 16.64 | 8.40 | 25.04 | 20.25 | 254.93 |
| | 512 | 0.4 | 55.56 | 11.05 | 7.66 | 18.71 | 10.13 | 230.59 |
| | 768 | 0.4 | 49.29 | 9.05 | 6.55 | 15.60 | 6.57 | 157.13 |
| | 1024 | 0.4 | 46.87 | 7.75 | 5.25 | 13.00 | 4.82 | 118.11 |
| | 0 | 0.5 | 62.89 | 30.84 | 9.75 | 40.59 | 43.56 | 550.36 |
| | 256 | 0.5 | 64.29 | 20.80 | 10.01 | 30.81 | 25.67 | 302.69 |
| | 512 | 0.5 | 62.44 | 13.49 | 8.89 | 22.38 | 12.70 | 250.36 |
| | 768 | 0.5 | 56.12 | 10.93 | 7.57 | 18.50 | 8.12 | 203.13 |
| | 1024 | 0.5 | 50.36 | 8.34 | 6.97 | 15.31 | 6.32 | 160.35 |
| | 0 | 0.6 | 70.35 | 33.17 | 10.20 | 43.37 | 45.86 | 553.07 |
| | 256 | 0.6 | 67.68 | 24.58 | 10.53 | 35.11 | 30.47 | 329.61 |
| | 512 | 0.6 | 63.74 | 16.23 | 9.45 | 25.68 | 15.63 | 266.98 |
| | 768 | 0.6 | 64.44 | 13.04 | 8.10 | 21.14 | 9.90 | 210.05 |
| | 1024 | 0.6 | 59.68 | 10.34 | 7.68 | 18.20 | 7.69 | 177.45 |
| | 0 | 0.7 | 62.35 | 35.36 | 9.76 | 45.12 | 48.08 | 573.42 |
| | 256 | 0.7 | 65.68 | 27.66 | 9.97 | 37.63 | 34.26 | 403.59 |
| | 512 | 0.7 | 68.64 | 18.88 | 9.37 | 28.25 | 18.43 | 283.76 |
| | 1024 | 0.7 | 64.68 | 12.93 | 6.90 | 19.83 | 8.25 | 209.12 |
| | 0 | 0.8 | 61.52 | 39.32 | 9.89 | 49.21 | 52.97 | 622.90 |
| | 256 | 0.8 | 66.37 | 30.67 | 9.70 | 40.37 | 41.66 | 489.61 |
| | 512 | 0.8 | 66.67 | 21.55 | 8.59 | 30.14 | 21.11 | 268.97 |
| | 1024 | 0.8 | 64.35 | 14.86 | 6.28 | 21.14 | 9.43 | 216.08 |
| Qwen3 | 0 | 0.4 | 56.89 | 28.44 | 18.30 | 46.74 | 41.08 | 594.94 |
| | 256 | 0.4 | 52.37 | 16.82 | 17.63 | 34.45 | 20.48 | 450.98 |
| | 512 | 0.4 | 58.57 | 11.03 | 17.00 | 28.03 | 10.11 | 421.74 |
| | 768 | 0.4 | 61.95 | 9.01 | 16.14 | 25.15 | 6.54 | 357.13 |
| | 1024 | 0.4 | 53.91 | 8.02 | 15.44 | 23.46 | 4.83 | 300.56 |
| | 0 | 0.6 | 69.56 | 34.90 | 20.26 | 55.16 | 48.63 | 642.10 |
| | 256 | 0.6 | 65.37 | 24.78 | 19.59 | 44.37 | 30.66 | 520.37 |
| | 512 | 0.6 | 66.57 | 16.22 | 19.50 | 35.72 | 15.40 | 471.09 |
| | 768 | 0.6 | 73.20 | 13.19 | 19.21 | 32.40 | 9.97 | 417.13 |
| | 1024 | 0.6 | 65.67 | 11.50 | 18.21 | 29.71 | 7.18 | 396.35 |
| | 0 | 0.8 | 67.68 | 37.97 | 20.18 | 58.15 | 50.81 | 759.15 |
| | 256 | 0.8 | 64.52 | 30.54 | 19.77 | 50.31 | 37.35 | 550.98 |
| | 512 | 0.8 | 67.37 | 21.35 | 19.36 | 40.71 | 20.98 | 461.02 |
| | 768 | 0.8 | 64.95 | 16.94 | 19.06 | 36.00 | 13.09 | 420.14 |
| | 1024 | 0.8 | 68.69 | 14.82 | 18.49 | 33.31 | 9.43 | 355.71 |

# E  PROMPTS

## E.1  LLM-AS-JUDGE

---

**Standard Tasks (Single-session-user/assistant Multi-session)**

I will give you a question, a correct answer, and a response from a model. Please answer yes if the response contains the correct answer. Otherwise, answer no. If the response is equivalent to the correct answer or contains all the intermediate steps to get the correct answer, you should also answer yes. If the response only contains a subset of the information required by the answer, answer no.
**Question:** {question}
**Correct Answer:** {answer}
**Model Response:** {response}
Is the model response correct? Answer yes or no only.

---

**Temporal Reasoning Tasks**

I will give you a question, a correct answer, and a response from a model. Please answer yes if the response contains the correct answer. Otherwise, answer no. If the response is equivalent to the correct answer or contains all the intermediate steps to get the correct answer, you should also answer yes. If the response only contains a subset of the information required by the answer, answer no. In addition, do not penalize off-by-one errors for the number of days. If the question asks for the number of days/weeks/months, etc., and the model makes off-by-one errors (e.g., predicting 19 days when the answer is 18), the model's response is still correct.
**Question:** {question}
**Correct Answer:** {answer}
**Model Response:** {response}
Is the model response correct? Answer yes or no only.

---

**Knowledge Update Tasks**

I will give you a question, a correct answer, and a response from a model. Please answer yes if the response contains the correct answer. Otherwise, answer no. If the response contains some previous information along with an updated answer, the response should be considered as correct as long as the updated answer is the required answer.
**Question:** {question}
**Correct Answer:** {answer}
**Model Response:** {response}
Is the model response correct? Answer yes or no only.

---

**Single-session Preference Tasks**

I will give you a question, a rubric for desired personalized response, and a response from a model. Please answer yes if the response satisfies the desired response. Otherwise, answer no. The model does not need to reflect all the points in the rubric. The response is correct as long as it recalls and utilizes the user's personal information correctly.
**Question:** {question}
**Rubric:** {answer}
**Model Response:** {response}
Is the model response correct? Answer yes or no only.

---

---

**Abstention Tasks**

I will give you an unanswerable question, an explanation, and a response from a model. Please answer yes if the model correctly identifies the question as unanswerable. The model could say that the information is incomplete, or some other information is given but the asked information is not.
**Question:** {question}
**Explanation:** {answer}
**Model Response:** {response}
Does the model correctly identify the question as unanswerable? Answer yes or no only.

---

