# OpenReview forum: "LightMem: Lightweight and Efficient Memory-Augmented Generation"
_ICLR.cc/2026/Conference — ICLR 2026 Poster_

### Official Review · Reviewer_SYFd · 2025-10-28

**Soundness:** 3
**Presentation:** 3
**Contribution:** 3
**Rating:** 8
**Confidence:** 3

**Summary:**

The paper proposes LightMem, a lightweight memory system for LLM agents that aims to keep long-term dialogue context both accurate and cheap to use. LightMem is explicitly inspired by the Atkinson–Shiffrin model of human memory, and contains: (1) a sensory memory module that pre-compresses and filters incoming dialogue with LLMLingua-2 and groups turns by topic, (2) a topic-aware short-term memory (STM) buffers these topic segments and summarises them adaptively rather than at fixed window boundaries, and (3) a long-term memory (LTM) that is updated in two stages — fast “soft updates” during inference and a later offline “sleep-time” consolidation.

The system is evaluated in an incremental multi-turn dialogue setting on LONGMEMEVAL-S, where the model only sees past turns as they arrive (simulating agents that operate over very long histories). On GPT-4o-mini and Qwen3-30B backbones, LightMem improves QA accuracy by up to ~10.9% over strong memory baselines such as A-Mem, and simultaneously cuts token usage by as much as 117×, reduces API calls by up to 177×, and lowers runtime by over 12×.

Ablations varying compression ratio and STM buffer threshold show a tradeoff between efficiency and accuracy but still indicate large gains over prior work across both backbones.

**Strengths:**

1. Clear, biologically grounded design. Structuring memory into sensory → STM → LTM with delayed consolidation is conceptually clean and directly motivated by cognitive models of human memory and sleep-driven consolidation, which is underexplored in LLM memory pipelines. Decoupling long-term memory maintenance from online inference and executing updates in parallel queues is a nice systems contribution. It directly addresses latency and consistency problems in prior approaches that update memories synchronously during inference.

2. Strong efficiency gains. The paper optimises for both higher QA accuracy as well as token cost, number of model calls, and wall-clock runtime. These are central bottlenecks for deployed LLM agents, and LightMem shows 1–2 orders of magnitude improvements on these metrics versus baselines like A-Mem, LangMem, MemoryOS, and Mem0.

3. Topic-aware STM. The idea of grouping turns by semantic/topic boundaries (via attention peaks and embedding similarity) instead of treating each turn or fixed window as a separate memory unit feels like a meaningful refinement over naive chunking and helps avoid both over-fragmented and over-mixed summaries.

4. Clarity. The paper is clearly written and easy to follow, with clear analysis/ablations of each component.

**Weaknesses:**

1. Benchmark breadth. All main results are on LONGMEMEVAL-S (500 long multi-session dialogues, ~110k tokens each on average) and QA-style evaluation judged by GPT-4o-mini. It’s not obvious how well LightMem generalizes to other domains (e.g., tool-use agents, planning tasks, code assistants) or to evaluation regimes that don’t rely on an LLM judge aligned with the backbone family.

2. Stability of offline updates. After “OP-update” (offline parallel update), accuracy sometimes drops relative to the purely online/soft-update configuration (e.g., GPT backbone at r=0.7 goes from 68.64% to 67.07%), suggesting that consolidation can introduce regressions. The paper claims improved long-term consistency, but the quantitative story is mixed and could use deeper analysis of failure modes.

3. Safety / faithfulness of summaries. The system stores summarized memory units as ground truth for future retrieval. We don’t see discussion of hallucinated summaries or factual drift when compressing many sessions into a single topic-level entry, which is a known risk for LLM-generated memory.

**Questions:**

1. Your evaluation is limited to LONGMEMEVAL-S and a QA-style setup with LLM-based judging. Can you provide evidence that LightMem generalizes beyond that setting? For example, to tool-using agents, planning/assistive tasks, or code assistants? If not, what are the expected failure modes outside the current benchmark?

2. When do you “sleep” the model? The LTM “sleep-time update” is described as happening offline in parallel and supposedly without user-facing latency. In a real interactive assistant, what triggers this (e.g. wall clock time, buffer size, explicit idle periods)? And what happens if the agent is in continuous use and never becomes idle?

3. Summary faithfulness audits. Have you measured factual faithfulness of STM summaries vs. gold dialogue histories (e.g., hallucination rate, omission rate)? Accuracy on QA is an indirect proxy but doesn’t capture whether LightMem invents or loses user-specific facts in ways that could create safety/privacy issues downstream.

---

> ### Author Response · Authors · 2025-11-23
> **Response to Reviewer SYFd [1/5]**
>
> Dear Reviewer SYFd,
>
> We are deeply grateful for your valuable time, great suggestion and affirmation. Below are our detailed responses to your concerns:
>
> ## **W1&Q1: Benchmark breadth.**
>
> Thank you very much for your insightful feedback. We have additionally incorporated two datasets: one is the most widely used memory benchmark, **LoCoMo**; furthermore, to demonstrate how well LightMem generalizes to other domains, we have selected the **ALFWorld** dataset. **ALFWorld** comprises household tasks, in each interaction round, the agent outputs an action, and the environment responds with textual feedback describing the resulting state. This process repeats for multiple turns until the task is completed or the maximum number of rounds is reached. ALFWorld includes test split to evaluate the agent’s generalization ability.
>
> **Results of LoCoMo**
>
> **backbone model: `gpt-4o-mini`, judge model: `gpt-4o-mini`.** (The results evaluated by other assessment methods are shown in another W1 below)
>
> | Method             | ACC(%) | Summary Tokens(k) In | Summary Tokens(k) Out | Update Tokens(k) In | Update Tokens(k) Out | Total Tokens(k) | Calls | Runtime(s) |
> |--------------------|--------|----------------------|-----------------------|---------------------|----------------------|-----------------|-------|------------|
> | FullText          | 71.83  | –                    | –                     | –                   | –                    | –               | –     | –          |
> | NaiveRAG          | 63.64  | –                    | –                     | –                   | –                    | –               | –     | –          |
> | LangMem           | 57.20  | –                    | –                     | 898.27              | 111.95               | 1010.22         | 920.62| 2229.37  |
> | A-MEM             | 64.16  | 182.74               | 49.29                 | 729.89              | 187.52               | 1149.43         | 1175.47| 6060.73  |
> | MemoryOS(locomo)    | 58.25  | 110.98               | 33.40                 | 78.08               | 64.54                | 287.00          | 553.45 | 2422.05  |
> | MemoryOS(normal)    | 54.87  | 226.86               | 46.61                 | 177.66              | 75.34                | 526.48          | 1016.06 | 3332.59  |
> | Mem0              | 61.69  | 851.32               | 10.53                 | 632.12              | 189.42                | 1683.39          | 1602.20 | 4432.87   |
> | LightMem(512,0.7) | 71.95  | 73.19                | 20.13                 | 6.05                | 0.40                 | 99.76           | 41.65   | 848.49    |
> | LightMem(768,0.7) | 70.26  | 57.54                | 18.92                 | 3.79                | 0.23                 | 80.48           | 29.55   | 737.80    |
> | LightMem(768,0.8) | 72.99  | 62.82                | 17.95                 | 4.14                | 0.28                 | 85.19           | 29.83  | 815.32     |
>
> **backbone model: `qwen3-30b-a3b-instruct-2507`, judge model: `gpt-4o-mini`.**
>
> | Method             | ACC(%) | Summary Tokens(k) In | Summary Tokens(k) Out | Update Tokens(k) In | Update Tokens(k) Out | Total Tokens(k) | Calls    | Runtime(s)  |
> | ------------------ | ------ | ---------- | ----------- | --------- | ---------- | ------------ | -------- | -------- |
> | FullText           | 74.87  | –          | –           | –         | –          | –            | –        | –        |
> | NaiveRAG           | 66.95  | –          | –           | –         | –          | –            | –        | –        |
> | LangMem            | 60.53  | –          | –           | 1004.35     | 138.02      | 1142.37       | 1005.37   | 2268.57   |
> | A-MEM              | 56.10  | 158.29     | 60.85       | 924.19    | 483.51     | 1,626.80     | 1175.40 | 5543.90 |
> | MemoryOS(locomo)   | 61.04  | 122.21     | 53.12       | 104.43    | 81.75      | 361.51       | 414.70   | 1269.70 |
> | MemoryOS(normal)   | 51.30  | 228.85     | 51.60       | 242.27    | 143.63     | 666.35       | 1004.60 | 1982.20 |
> | Mem0               | 43.31  | 827.09     | 18.64       | 763.88    | 189.80     | 1799.40     | 1614.50 | 4540.70 |
> | LightMem(768,0.6)  | 71.36  | 56.68      | 34.14       | 8.31      | 0.74       | 99.87        | 29.10    | 815.70   |
> | LightMem(1024,0.8) | 72.60  | 61.38      | 36.33       | 9.86      | 0.88       | 108.45       | 32.00    | 1079.40 |

---

> ### Author Response · Authors · 2025-11-23
> **Response to Reviewer SYFd [2/5]**
>
> > Note: MemoryOS(locomo) in the table refers to a script in the MemoryOS library specifically optimized for the LoCoMo dataset. This script significantly simplifies the regular version of MemoryOS. Therefore, we also tested the regular version of MemoryOS, labeled as MemoryOS(regular) in the table.
>
> > Note: For better readability, we have consolidated the original two rows of LightMem results into a single row, and the reported ACC values are based on the results after offline updates.
>
> **Conclusions:**
>
> (1) For GPT backbone, it can be observed that LightMem outperforms other memory baseline methods in **ACC by 6.10% to 18.12%**, while achieving a **2.87x to 20.92x improvement** in total token efficiency, a **13.29x to 39.78x reduction** in API calls, and a **2.63x to 8.21x speedup** in total runtime compared to other baselines.
>
> (2) It can be observed that, on the Qwen backbone, **LightMem continues to outperform all other baselines in both effectiveness and efficiency**. Specifically, it achieves **4.41%–29.29% higher ACC**, delivers a **3.33× to 18.02× reduction in total token consumption**, reduces **API call count by 12.96× to 55.48×**, and provides **1.18× to 5.57× speed-ups in runtime**.
>
> **This demonstrates the generalizability of our method.**
>
> **Results of ALFWorld**
>
>
> | Model             | Granularity       | ALFWorld Dev ↑ | ALFWorld Test ↑ | ALFWorld Steps ↓ |
> |-------------------|-------------------|----------------|-----------------|------------------|
> | GPT-4o            | No Memory         | 39.28          | 42.14           | 23.76           |
> |                   | Raw Trajectory    | 67.17          | 74.29           | 16.49            |
> |                   | LightMem          | 77.90          | 69.42           | 14.66            |
> | Qwen2.5-72b       | No Memory         | 44.91          | 41.25           | 21.38            |
> |                   | Raw Trajectory    | 64.49          | 69.57           | 16.40            |
> |                   | LightMem          | 83.24          | 73.20           | 15.02            |
>
> > Since this dataset is not a traditional memory benchmark, adapting it to fit other memory baselines of this paper is highly time-consuming. Therefore, due to time constraints, we have only selected the following baselines:
> (1) **No Memory**: Directly testing the backbone model on the test set;
> (2) **Raw Trajectory**: At inference stage, the system identifies the top-k trajectories with the highest semantic similarity to the current task based on their vector representations, and utilizes them as procedural memories to guide subsequent execution. These trajectories are pre-collected gold instances filtered from the training set.
>
> > **LightMem**: During the training phase, we store the entire trajectory using LightMem. We adapted the prompt used by `f_{sum}()` for this dataset to extract abstract guidelines and generalized experiences from the raw trajectories in the STM buffer. These are then stored in the long-term memory alongside the original trajectories. During testing, the LTM is utilized for inference.
>
> > Dev and Test are two evaluation sets, where the task types in Dev are relatively similar to those in the training set, while Test contains entirely new tasks.
> There are 140 test samples and 134 test samples, respectively.
>
> As observed, after integrating LightMem, task performance has improved on both base models while the number of decision steps has decreased, demonstrating the feasibility of agentifying this paradigm.

---

> ### Author Response · Authors · 2025-11-23
> **Response to Reviewer SYFd [3/5]**
>
> ## **W1: Potential Evaluation Bias**
>
> Thank you for your valuable suggestion. We have conducted the following experiments to ensure the reliability of the LLM-as-Judge evaluation results and minimize potential biases arising from using models within the same family.
>
> (1) On the LoCoMo dataset, in addition to using `GPT-4o-mini` itself for evaluation, we also employed a model from a different family, `Qwen2.5-32B-Instruct`, to assess the obtained results. The experimental findings are as follows:
>
> | Method | Judge Model | Overall ↑ | Single Hop | Multi Hop | Open Domain | Temporal |
> | :--- | :--- | :---: | :---: | :---: | :---: | :---: |
> | FullText         | gpt-4o-mini | 73.83 | 68.79 | 56.25 | 86.56 | 50.16 |
> |                  | qwen2.5-32b-instruct | 73.18 | 68.09 | 54.17 | 86.21 | 49.22 |
> | NaiveRAG         | gpt-4o-mini | 63.64 | 55.32 | 47.92 | 70.99 | 56.39 |
> |                  | qwen2.5-32b-instruct | 63.12 | 53.55 | 50.00 | 71.34 | 53.89 |
> | LangMem          | gpt-4o-mini | 57.20 | 52.10 | 41.65 | 62.80 | 43.25 |
> |                  | qwen2.5-32b-instruct | 56.80 | 51.45 | 42.30 | 63.75 | 42.90 |
> | A-MEM            | gpt-4o-mini | 64.16 | 56.03 | 31.25 | 72.06 | 60.44 |
> |                  | qwen2.5-32b-instruct | 60.71 | 53.55 | 32.29 | 69.08 | 53.58 |
> | MemoryOS(eval)   | gpt-4o-mini | 58.25 | 56.74 | 45.83 | 67.06 | 40.19 |
> |                  | qwen2.5-32b-instruct | 61.04 | 64.18 | 40.62 | 70.15 | 40.50 |
> | MemoryOS(pypi)   | gpt-4o-mini | 54.87 | 52.13 | 43.75 | 63.97 | 36.76 |
> |                  | qwen2.5-32b-instruct | 55.91 | 52.48 | 41.67 | 66.35 | 35.83 |
> | Mem0             | gpt-4o-mini | 61.69 | 56.38 | 43.75 | 66.47 | 59.19 |
> |                  | qwen2.5-32b-instruct | 61.69 | 54.26 | 46.88 | 67.66 | 57.01 |
> | LightMem(512,0.7)| gpt-4o-mini | 71.95 | 62.41 | 44.79 | 77.41 | 74.14 |
> |                  | qwen2.5-32b-instruct | 73.90 | 69.15 | 50.00 | 78.00 | 74.45 |
> | LightMem(768,0.7)| gpt-4o-mini | 70.26 | 62.06 | 42.71 | 74.67 | 74.14 |
> |                  | qwen2.5-32b-instruct | 72.40 | 64.54 | 43.75 | 77.17 | 75.39 |
> | LightMem(768,0.8)| gpt-4o-mini | 72.99 | 67.02 | 45.83 | 76.81 | 76.32 |
> |                  | qwen2.5-32b-instruct | 74.35 | 68.79 | 47.92 | 78.24 | 76.95 |
>
> (2) On the LoCoMo dataset, in addition to using the `LLM-as-a-judge` paradigm, we also introduced `human evaluation`. We randomly sampled 250 questions according to the proportional distribution of the four question types, and the results are as follows:
>
> | Method | Evaluation Method | Overall ↑ | Single | Multi | Open | Temp |
> | :--- | :--- | :---: | :---: | :---: | :---: | :---: |
> | FullText         | gpt-4o-mini | 70.80 | 64.15 | 33.33 | 84.33 | 52.08 |
> |                  | human evaluation | 67.20 | 56.72 | 28.67 | 82.60 | 47.83 |
> | NaiveRAG         | gpt-4o-mini | 64.80 | 56.52 | 40.00 | 73.08 | 59.32 |
> |                  | human evaluation | 65.80 | 54.00 | 37.33 | 74.00 | 59.93 |
> | A-MEM            | gpt-4o-mini | 60.40 | 60.87 | 20.00 | 65.38 | 62.96 |
> |                  | human evaluation | 56.20 | 39.96 | 33.00 | 68.38 | 49.30 |
> | MemoryOS(eval)   | gpt-4o-mini | 58.40 | 53.19 | 42.86 | 69.47 | 41.38 |
> |                  | human evaluation | 58.00 | 44.30 | 48.86 | 73.94 | 35.31 |
> | MemoryOS(pypi)   | gpt-4o-mini | 48.00 | 35.42 | 29.41 | 56.82 | 43.40 |
> |                  | human evaluation | 44.80 | 31.25 | 29.41 | 54.55 | 37.74 |
> | Mem0             | gpt-4o-mini | 63.20 | 64.86 | 55.56 | 63.08 | 66.07 |
> |                  | human evaluation | 65.60 | 53.24 | 65.56 | 70.77 | 61.79 |
>
> It can be observed that whether using an LLM judge from a `different model family` or introducing `human evaluation`, the final assessment results show **minimal deviation**. This demonstrates the consistency among these evaluation methods. We sincerely appreciate your suggestion and will incorporate all these evaluation approaches and results into the updated version of our paper.

---

> ### Author Response · Authors · 2025-11-23
> **Response to Reviewer SYFd [4/5]**
>
> ## **W2:Stability of offline updates.**
> Thank you for your insightful suggestion. Indeed, occasional performance fluctuations can occur after offline parallel updates, which is a challenge faced by all current memory systems. During the update process, the mainstream approach is to directly concatenate memory entries into prompts and invoke the backbone model to determine whether updates are needed and take corresponding actions (e.g., delete, update, merge, ignore). However, due to semantic complexities, the model may occasionally take incorrect actions, as mentioned in the case study of our paper. This highlights the need for further exploration of better memory update mechanisms.
>
> Regarding your suggestion for deeper quantitative research, as shown in the Table1 of main paper, the performance degradation after offline parallel updates is minimal, resulting in a very small sample size available for analysis (likely only 4-5 samples). We apologize that we are currently unable to conduct more in-depth quantitative studies on this issue. Ultimately, however, the performance decline after updates largely stems from the heavy reliance of current memory system update mechanisms on the backbone model.
>
> ## **W3&Q3:Safety / faithfulness of summaries.**
>
> Thank you for the very insightful feedback.
>
> First, we would like to clarify a potential misunderstanding: LightMem does **not** always perform *“compressing many sessions into a single topic-level entry.”* As described in Section 3.2,
> $
> \text{sum}_i = f_{\text{sum}}(S_i),\quad
> S_i \subseteq \{\text{user}_i,\ \text{model}_i\},\quad
> S_i \neq \varnothing.
> $
> Here, you can see, $S_i$ is defined at the **turn level**, meaning each turn is summarized individually. In practice, this corresponds to extremely fine-grained, near turn-level summarization (though occasionally a few consecutive turns may be summarized together). Therefore, LightMem does *not* store “a single topic-level entry,” but rather **multiple topic-aware memory entries**. The “topic” merely provides a semantic boundary that helps the LLM summarize more accurately; each topic segment still produces *multiple* summaries rather than one coarse entry.
>
> Because summarization is highly fine-grained, factual details, especially event-level information, tend to be preserved more reliably, and retrieval also benefits from this granularity. However, we agree with you that hallucinated summaries and factual drift can still occur. Indeed, we observed such cases during our experiments. For example:
>
> Q: How was Melanie’s son feeling during the accident, and how did he handle it?
>
> A: He was scared but reassured by his family
>
> **Evidence turn:**
> Caroline: “The kids look so cute, Mel! I bet they bring lots of joy. How did they handle the accident?”
> Melanie: “Thanks! They were scared but we reassured them and explained their brother would be OK. They're tough kids.”
>
> **The corresponding input to LightMem’s summarization module (after pre compression) is:**
> Caroline: “kids cute, Mel! joy. handle accident?”
> Melanie: “scared reassured explained their brother OK. tough kids.”
>
> **LightMem summary produced:**
> “Melanie reassured her children after the accident, explaining that their brother is okay, and they are tough kids.”
>
> In this case, although “scared” is present in the compressed input, the summarizer omitted the children were "scared”, which later caused an incorrect QA response (“He is okay.”). This example shows that hallucination or detail loss can indeed happen.
>
> We believe this issue is influenced by both prompt design and the capability of the underlying base model. Notably, several recent works focus on this problem:
> - **MEM-α** [1], which treats summarization as a memory-management tool and uses RL to train the model to produce more faithful summaries;
> - **HalluMem** [2], a benchmark designed to diagnose and localize memory-induced hallucinations, helping prevent error accumulation and factual drift.
>
> We sincerely appreciate the reviewer for highlighting this important concern. We will incorporate the above discussion, along with the two related papers, into our revised manuscript, and we also plan to **conduct a systematic evaluation of LightMem on HalluMem in future work**.
>
> [1] Mem-α: Learning Memory Construction via Reinforcement Learning. https://arxiv.org/abs/2509.25911
>
> [2] HaluMem: Evaluating Hallucinations in Memory Systems of Agents. https://arxiv.org/abs/2511.03506

---

> ### Author Response · Authors · 2025-11-23
> **Response to Reviewer SYFd [5/5]**
>
> ## **Q2:When do you “sleep” the model?**
>
> Thank you again for raising such a professional and insightful question. My response is as follows:
>
> At the current stage, LightMem still relies on a manually triggered offline update process. In our experimental pipeline, after storing the entire long dialogue into the memory pool (with soft updates performed online), we then execute the offline update in a batch manner. The performance of LightMem is evaluated both **before** and **after** this offline update.
>
> In real-world scenarios, however, the triggering mechanism should be customized based on application requirements. For example, wall-clock–based triggers are useful in settings that demand high real-time accuracy, where outdated information should be periodically removed or deprioritized during retrieval. Event-driven triggers are also feasible—for instance, if the model detects a strong conflict between a new entry and existing entries, an immediate update may be necessary. This naturally suggests combining **online** and **offline** updates. Moreover, LightMem could potentially be integrated with reinforcement learning to learn an automated update-triggering policy, enabling the backbone model to determine *when* and *how* offline updates should be executed.
>
> **Regarding the question “What happens if the agent is in continuous use and never becomes idle?”**
>
> When an offline update is triggered, the system can create a snapshot of the memory at that moment and perform the offline update on the snapshot. Meanwhile, the continuously operating agent continues to use and softly update the original memory pool, ensuring uninterrupted access to global memory. The softly updated entries produced during this period can be marked, and once the offline update completes, the updated memory pool can be merged with these marked entries to achieve incremental consistency.
>
> Thank you again for your highly professional question, it has provided us with valuable insights.
>
>
> **Please let us know if you have any further questions. If you find that our response addresses some of your concerns, would you kindly consider raising your rating score for our paper? We greatly appreciate your consideration.**
>
> Best Regards,
>
> Authors of LightMem

---

> ### Author Response · Authors · 2025-11-27
> **Additional Clarifications and Updates for Reviewer SYFd**
>
> Dear Reviewer SYFd,
>
> We highly appreciate the constructive comments and insightful suggestions you have offered for our work.
>
> Additionally, compared with our initial submission, we have now included the performance of the **Qwen backbone model** on the **LoCoMo** dataset, with the results presented above.
>
> As the discussion period has only a few days left, in order for us to have sufficient time to address any additional questions you may have, we kindly encourage you to engage in the ongoing discussion and share any further insights or clarifications you may have.
>
> If you find that our response addresses some of your concerns, would you kindly consider raising your rating score for our paper?
>
> Thank you very much for your time and consideration. We look forward to hearing from you soon.
>
> Best Regards,
>
> Authors of LightMem

---

> ### Comment · Reviewer_SYFd · 2025-11-27
> **Response to Authors**
>
> Dear authors,
>
> Thank you for your detailed responses, and for clarification on the above concerns. I am satisfied with your answers, and think that your response to my question about "when to sleep" the model is particularly interesting - I recommend that you include a brief discussion of this notion in your future work. All in all I am happy to maintain my positive score.

---

> ### Author Response · Authors · 2025-11-27
> **Appreciation Note to Reviewer SYFd**
>
> Dear Reviewer SYFd,
>
> Thank you very much for your helpful suggestions and for maintaining a positive score. We truly appreciate your insight regarding the “when to sleep” notion, and we will be sure to include a discussion of this point in the future work. If you have any additional questions or concerns, please feel free to let us know, we will do our utmost to address them.
>
> Best regards,
>
> Authors of LightMem

---

### Official Review · Reviewer_4xu7 · 2025-10-31

**Soundness:** 3
**Presentation:** 3
**Contribution:** 3
**Rating:** 6
**Confidence:** 1

**Summary:**

This paper presents LightMem, a lightweight memory system for large language models (LLMs) designed to improve long-context reasoning in multi-turn dialogues while minimizing inference and maintenance overhead. Inspired by the Atkinson–Shiffrin model of human memory, LightMem consists of three stages: (1) a pre-compression sensory memory that filters redundant tokens using LLMLingua-2, (2) a topic-aware short-term memory (STM) that segments and summarizes dialogue by semantic topics, and (3) a sleep-time long-term memory (LTM) update mechanism that asynchronously consolidates memory to reduce runtime latency.

**Strengths:**

1. The decomposition of memory into sensory / STM / LTM stages reflects cognitive plausibility, offering interpretability and modularity.
2. Across GPT and Qwen backbones, LightMem consistently outperforms prior methods in accuracy, latency, API usage, and token efficiency.
3. Employing LLMLingua-2 for token-level retention filtering is well-grounded and shown to maintain semantic fidelity even at 50–70% compression ratios.
4. Writing is clear, figures and tables are well-structured, and Appendix contains implementation details, metrics, and additional experiments.

**Weaknesses:**

1. All evaluation is performed on the QA-focused LONGMEMEVAL benchmark. It remains unclear how LightMem would perform in generation-based tasks, e.g., reasoning or question-answering.

**Questions:**

1. How well LightMem scales over long durations of deployment, such as weeks- or months-long open-ended interactions? Does the framework include mechanisms to manage memory lifecycle over time to avoid memory bloat and coherence degradation?

---

> ### Author Response · Authors · 2025-11-23
> **Response to Reviewer 4xu7 [1/3]**
>
> Dear Reviewer 4xu7,
>
> We are deeply grateful for your valuable time, great suggestion and affirmation. Below are our detailed responses to your concerns:
>
> ## **W1: Benchmark breadth.**
>
> Thank you very much for your insightful feedback. We have additionally incorporated two datasets: one is the most widely used memory benchmark, **LoCoMo**; furthermore, to demonstrate how well LightMem generalizes to other domains, we have selected the **ALFWorld** dataset. **ALFWorld** comprises household tasks, in each interaction round, the agent outputs an action, and the environment responds with textual feedback describing the resulting state. This process repeats for multiple turns until the task is completed or the maximum number of rounds is reached. ALFWorld includes test split to evaluate the agent’s generalization ability.
>
> **Results of LoCoMo**
>
> **backbone model: `gpt-4o-mini`, judge model: `gpt-4o-mini`.**
>
> | Method             | ACC(%) | Summary Tokens(k) In | Summary Tokens(k) Out | Update Tokens(k) In | Update Tokens(k) Out | Total Tokens(k) | Calls | Runtime(s) |
> |--------------------|--------|----------------------|-----------------------|---------------------|----------------------|-----------------|-------|------------|
> | FullText          | 71.83  | –                    | –                     | –                   | –                    | –               | –     | –          |
> | NaiveRAG          | 63.64  | –                    | –                     | –                   | –                    | –               | –     | –          |
> | LangMem           | 57.20  | –                    | –                     | 898.27              | 111.95               | 1010.22         | 920.62| 2229.37  |
> | A-MEM             | 64.16  | 182.74               | 49.29                 | 729.89              | 187.52               | 1149.43         | 1175.47| 6060.73  |
> | MemoryOS(locomo)    | 58.25  | 110.98               | 33.40                 | 78.08               | 64.54                | 287.00          | 553.45 | 2422.05  |
> | MemoryOS(normal)    | 54.87  | 226.86               | 46.61                 | 177.66              | 75.34                | 526.48          | 1016.06 | 3332.59  |
> | Mem0              | 61.69  | 851.32               | 10.53                 | 632.12              | 189.42                | 1683.39          | 1602.20 | 4432.87   |
> | LightMem(512,0.7) | 71.95  | 73.19                | 20.13                 | 6.05                | 0.40                 | 99.76           | 41.65   | 848.49    |
> | LightMem(768,0.7) | 70.26  | 57.54                | 18.92                 | 3.79                | 0.23                 | 80.48           | 29.55   | 737.80    |
> | LightMem(768,0.8) | 72.99  | 62.82                | 17.95                 | 4.14                | 0.28                 | 85.19           | 29.83  | 815.32     |
>
> **backbone model: `qwen3-30b-a3b-instruct-2507`, judge model: `gpt-4o-mini`.**
>
> | Method             | ACC(%) | Summary Tokens(k) In | Summary Tokens(k) Out | Update Tokens(k) In | Update Tokens(k) Out | Total Tokens(k) | Calls    | Runtime(s)  |
> | ------------------ | ------ | ---------- | ----------- | --------- | ---------- | ------------ | -------- | -------- |
> | FullText           | 74.87  | –          | –           | –         | –          | –            | –        | –        |
> | NaiveRAG           | 66.95  | –          | –           | –         | –          | –            | –        | –        |
> | LangMem            | 60.53  | –          | –           | 1004.35     | 138.02      | 1142.37       | 1005.37   | 2268.57   |
> | A-MEM              | 56.10  | 158.29     | 60.85       | 924.19    | 483.51     | 1,626.80     | 1175.40 | 5543.90 |
> | MemoryOS(locomo)   | 61.04  | 122.21     | 53.12       | 104.43    | 81.75      | 361.51       | 414.70   | 1269.70 |
> | MemoryOS(normal)   | 51.30  | 228.85     | 51.60       | 242.27    | 143.63     | 666.35       | 1004.60 | 1982.20 |
> | Mem0               | 43.31  | 827.09     | 18.64       | 763.88    | 189.80     | 1799.40     | 1614.50 | 4540.70 |
> | LightMem(768,0.6)  | 71.36  | 56.68      | 34.14       | 8.31      | 0.74       | 99.87        | 29.10    | 815.70   |
> | LightMem(1024,0.8) | 72.60  | 61.38      | 36.33       | 9.86      | 0.88       | 108.45       | 32.00    | 1079.40 |

---

> ### Author Response · Authors · 2025-11-23
> **Response to Reviewer 4xu7 [2/3]**
>
> > Note: MemoryOS(locomo) in the table refers to a script in the MemoryOS library specifically optimized for the LoCoMo dataset. This script significantly simplifies the regular version of MemoryOS. Therefore, we also tested the regular version of MemoryOS, labeled as MemoryOS(regular) in the table.
>
> > Note: For better readability, we have consolidated the original two rows of LightMem results into a single row, and the reported ACC values are based on the results after offline updates.
>
> **Conclusions:**
>
> (1) For GPT backbone, it can be observed that LightMem outperforms other memory baseline methods in **ACC by 6.10% to 18.12%**, while achieving a **2.87x to 20.92x improvement** in total token efficiency, a **13.29x to 39.78x reduction** in API calls, and a **2.63x to 8.21x speedup** in total runtime compared to other baselines.
>
> (2) It can be observed that, on the Qwen backbone, **LightMem continues to outperform all other baselines in both effectiveness and efficiency**. Specifically, it achieves **4.41%–29.29% higher ACC**, delivers a **3.33× to 18.02× reduction in total token consumption**, reduces **API call count by 12.96× to 55.48×**, and provides **1.18× to 5.57× speed-ups in runtime**.
>
> **This demonstrates the generalizability of our method.**
>
> **Results of ALFWorld**
>
>
> | Model             | Granularity       | ALFWorld Dev ↑ | ALFWorld Test ↑ | ALFWorld Steps ↓ |
> |-------------------|-------------------|----------------|-----------------|------------------|
> | GPT-4o            | No Memory         | 39.28          | 42.14           | 23.76           |
> |                   | Raw Trajectory    | 67.17          | 74.29           | 16.49            |
> |                   | LightMem          | 77.90          | 69.42           | 14.66            |
> | Qwen2.5-72b       | No Memory         | 44.91          | 41.25           | 21.38            |
> |                   | Raw Trajectory    | 64.49          | 69.57           | 16.40            |
> |                   | LightMem          | 83.24          | 73.20           | 15.02            |
>
> > Since this dataset is not a traditional memory benchmark, adapting it to fit other memory baselines of this paper is highly time-consuming. Therefore, due to time constraints, we have only selected the following baselines:
> (1) **No Memory**: Directly testing the backbone model on the test set;
> (2) **Raw Trajectory**: At inference stage, the system identifies the top-k trajectories with the highest semantic similarity to the current task based on their vector representations, and utilizes them as procedural memories to guide subsequent execution. These trajectories are pre-collected gold instances filtered from the training set.
>
> > **LightMem**: During the training phase, we store the entire trajectory using LightMem. We adapted the prompt used by `f_{sum}()` for this dataset to extract abstract guidelines and generalized experiences from the raw trajectories in the STM buffer. These are then stored in the long-term memory alongside the original trajectories. During testing, the LTM is utilized for inference.
>
> > Dev and Test are two evaluation sets, where the task types in Dev are relatively similar to those in the training set, while Test contains entirely new tasks.
> There are 140 test samples and 134 test samples, respectively.
>
> As observed, after integrating LightMem, task performance has improved on both base models while the number of decision steps has decreased, demonstrating the feasibility of agentifying this paradigm.

---

> ### Author Response · Authors · 2025-11-23
> **Response to Reviewer 4xu7 [3/3]**
>
> ## **Q1: How well LightMem scales over long durations of deployment?**
>
> Thank you for your question, it's a very practical one. I believe our main experiments already shed some light on this issue. The dataset used in our primary experiments, **LongMemEval**, is inherently lengthy, containing 500 dialogue trajectories with each trajectory averaging nearly **110K tokens**. We selected this dataset precisely because its interaction trajectories are sufficiently long, making it more representative of real-world scenarios. The experimental results demonstrate that LightMem outperforms other baselines in both **effectiveness** and **efficiency**, proving its superior performance in long-interaction scenarios.
>
> As for **very long-term interactions**, such as open-ended interactions spanning weeks or months of deployment, this could indeed lead to an extremely large memory bank and a potential decline in retrieval accuracy due to the bloated memory structure. This scenario remains an open area for exploration. However, based on the current results, I believe LightMem still holds greater potential in addressing these challenges.
>
> ## **Q2: Does the framework include mechanisms to manage memory lifecycle over time?**
>
> Thank you for your excellent question. Currently, LightMem does not incorporate such a mechanism, but I agree it is necessary. In scenarios involving very long-term interactions, it is essential to design mechanisms to manage the memory lifecycle. This could involve implementing a hierarchical memory structure, incorporating metrics for importance and timeliness, or other similar strategies. We will continue to refine LightMem's design in subsequent work.
>
> Thank you again for your constructive suggestions!
>
> **Please let us know if you have any further questions. If you find that our response addresses some of your concerns, would you kindly consider raising your rating score for our paper? We greatly appreciate your consideration.**
>
> Best Regards,
>
> Authors of LightMem

---

> > ### Comment · Reviewer_4xu7 · 2025-11-26
> >
> > Thank the authors for the clarifications provided.

---

> > > ### Author Response · Authors · 2025-11-27
> > > **Additional Clarifications and Updates for Reviewer 4xu7**
> > >
> > > Dear Reviewer 4xu7,
> > >
> > > We are sincerely grateful for your positive feedback. We appreciate the time you took to review our rebuttal.
> > >
> > > Additionally, compared with our initial submission, we have now included the performance of the **Qwen backbone model** on the **LoCoMo** dataset, with the results presented above. We would be truly grateful if you could take a moment to review these updated findings. If you have any further questions or concerns, please feel free to let us know, we remain fully committed to addressing them with our utmost effort.
> > >
> > > We will ensure that all the discussed clarifications and improvements are incorporated into the final version of the paper.
> > >
> > > Best Regards,
> > >
> > > Authors of LightMem

---

### Official Review · Reviewer_Ub6P · 2025-11-01

**Soundness:** 3
**Presentation:** 3
**Contribution:** 3
**Rating:** 6
**Confidence:** 3

**Summary:**

The paper describes LightMem, a memory module for LLMs, inspired by a model of the human memory. It has three stages: (1) filtering incoming tokens which are of low quality or redundant, (2) a short-term memory which groups together semantically related utterances, and (3) a long-term memory which is maintained during designated offline periods where memories are re-organized, removing redundancies and resolving inconsistencies. By moving this heavy lifting offline, LightMem ensures that the processes involved with maintaining the memory have little impact on the LLM’s latency. Experiments show significant improvement over baselines in terms of accuracy, number of tokens, number of API calls, and runtime.

**Strengths:**

By and large, the paper is well written and well motivated. The stages of creating and generating memory entries are inspired by the model of the human brain. The experimental setup is sufficient and results are convincing. The field of memory-augmented LLMs is crowded. While the paper does not present fundamentally different ideas to the ones in previous work, I still find it interesting and I think that it is a solid addition to this growing body of work.

**Weaknesses:**

I don’t have any major issues with the paper. The description in Section 3.3 is not clear to me. What happens after a queue for each entry is created? How is f_{update} implemented? What happens when the LTM reaches capacity?

**Questions:**

N/A

---

> ### Author Response · Authors · 2025-11-23
> **Response to Reviewer Ub6P [1/1]**
>
> Dear Reviewer Ub6P,
>
> We are deeply grateful for your valuable time, great suggestion and affirmation. Below are our detailed responses to your concerns:
>
> ## **W1: Clarification Needed on Memory Update Mechanism.**
>
> Thank you for pointing out the details in our paper. Below, I will provide a detailed explanation of LightMem's offline update mechanism, and we will refine and incorporate this content into the paper.
>
> First, let's clarify the update relationship between each entry and its update queue: the current entry is the target of the update, while the entries in its update queue have the potential to update it. For example, if there is a conflict between the entries in the update queue and the current entry, the current entry will be deleted.
>
> Second, `f_update()` is implemented using the LLM backbone in this paper. Following the update mechanisms of existing memory-related work, we concatenate the entry and the potential entries that may update it (retrieved via vector search with timestamp constraints) into a prompt. We then design the prompt to provide `f_update()` with several operations—such as delete, update, and merge—and finally, let `f_update()` identify semantic conflicts and output a specific operation.
>
> In the offline update process of LightMem: each entry maintains an update queue. The current entry and the entries in its update queue are concatenated into a prompt, and the LLM is called to determine the operation to be performed on the current entry, such as updating, deleting, or merging. The advantage of this approach is that it enables parallelization of the entire offline update process. Since each entry maintains its own update queue and the update operation is limited to the current entry, it avoids forming complex update relationships across the entire memory bank (e.g., A updates B, and B updates C). For example, if there are 500 entries in the memory bank, 500 update queues will be created. The `f_update()` function can then make decisions in parallel, with each entry and its update queue producing only one operation. Once all 500 operation results are obtained, they are executed together—deleting or merging entries as needed—to update the entire memory bank.
>
> Finally, the LTM does not have a capacity limit. Since the long-term memory is the final memory bank, it is typically implemented using a vector database or knowledge graph. In all experiments in this paper, the LTM is implemented as a vector database. As long as there is sufficient disk space, it is unlikely for the LTM to reach its capacity limit.
>
> Thank you again for your constructive suggestions!
>
> **Please let us know if you have any further questions. If you find that our response addresses some of your concerns, would you kindly consider raising your rating score for our paper? We greatly appreciate your consideration.**
>
> Best Regards,
>
> Authors of LightMem

---

> > ### Comment · Reviewer_Ub6P · 2025-11-26
> >
> > Thank you for the clarifications.

---

> > > ### Author Response · Authors · 2025-11-27
> > > **Appreciation Note to Reviewer Ub6P**
> > >
> > > Dear Reviewer Ub6P,
> > >
> > > We are sincerely grateful for your positive feedback. We appreciate the time you took to review our rebuttal. We will ensure that all the discussed clarifications and improvements are incorporated into the final version of the paper. If you have any further questions or concerns, please feel free to let us know, we remain fully committed to addressing them with our utmost effort.
> > >
> > > Best Regards,
> > >
> > > Authors of LightMem

---

### Official Review · Reviewer_sGUo · 2025-11-01

**Soundness:** 3
**Presentation:** 3
**Contribution:** 3
**Rating:** 6
**Confidence:** 2

**Summary:**

This paper proposes LightMem, a memory system approach that augments LLMs to better retain and retrieve information over long conversations. LightMem, inspired by Atkinson-Shiffrin's model of human memory, consists 3 components: (1) a sensory memory that filters out noise and retain only the most informative parts, (2) a topic aware short term memory that consolidates information for more structed access, and (3) a long term memory that that decouples consolidation from online inference. The authors report significant gains on several metrics using LightMem on GPT and Qwen models.

**Strengths:**

* I think this is decent engineering effort to build memory augmented LLMs.

**Weaknesses:**

* I am not sure about the novelty of this work. The base idea seems to have been studied in prior works with different names/flavors.

**Questions:**

1. I am not really an expert in the memory-augmented LLMs area, but with cursory google search I found [He et al, 2025](https://arxiv.org/pdf/2405.06067) and [Sun and Zeng, 2025](https://arxiv.org/abs/2507.22925), both of them seem very relevant to this work. You should include a discussion how your approach differ from these previous works, possibly also include them as a baseline.

2. The paper might be missing some important citations such as [Wu et al, 2025](https://arxiv.org/pdf/2504.15965) that has some discussion on memory systems in humans vs LLMs.

---

> ### Author Response · Authors · 2025-11-23
> **Response to Reviewer sGUo [1/2]**
>
> ear Reviewer sGUo,
>
> We are deeply grateful for your valuable time and affirmation. Below are our detailed responses to your concerns:
>
> ## **W1: Clarification on Contribution Positioning.**
>
> In the field of RAG/memory, there are also works inspired by the human brain or cognitive science, such as HippoRAG [1] and RoboMemory [2]. However, to the best of my knowledge, our work is currently the only one that extensively mimics and leverages the lightweight characteristics of these human brain mechanisms, such as using sensory memory to filter initial redundant information and updating long-term memory during sleep time.
>
> [1] HippoRAG: Neurobiologically Inspired Long-Term Memory for Large Language Models. NeurIPS 2024
>
> [2] RoboMemory: A Brain-inspired Multi-memory Agentic Framework for Interactive Environmental Learning in Physical Embodied Systems. https://arxiv.org/abs/2508.01415
>
> ## **Q1: Consider discussing the distinctions between your approach and recent relevant works (e.g., He et al., 2025; Sun & Zeng, 2025).**
>
> Thank you for pointing out these two references and suggesting we include a discussion of them.
>
> We have carefully reviewed _HMT: Hierarchical Memory Transformer for Efficient Long-Context Language Processing_ [1] and _H-MEM: Hierarchical Memory for High-Efficiency Long-Term Reasoning in LLM Agents_ [2]. We appreciate the suggestion. Interestingly, these two papers naturally fall into two distinct categories in the taxonomy of memory systems. Below we clarify how our approach differs from both, from the perspectives of memory representation and efficiency optimization.
>
> **Memory Representation**
>
> From the perspective of **memory representation**, memory systems can be divided into **Parametric Memory Systems** and **Non-parametric Memory Systems**. Parametric memory systems refer to approaches where information is stored directly within model parameters. A seminal contribution in this direction is the Long Short-Term Memory (LSTM) architecture [3], which represents memory using a hidden state vector. Through training, neural networks learn to update this hidden state and retrieve information from it. However, the representational capacity of such hidden vectors is inherently limited. To address this, subsequent studies have expanded memory size [4-8] or introduced more sophisticated update mechanisms [9-12] to improve parametric memory.
>
> **HMT** falls into this **parametric** category. It enhances both the representational capacity of the hidden state and the mechanism used to update it. Despite these advances, parametric memory faces two persistent challenges [13]. First, **it lacks interpretability**, as knowledge is entangled within learnable parameters. Second, **it is difficult to modify**, even with progress in knowledge editing [14-15] and machine unlearning techniques [16-17]. In addition, many parametric memory systems including **HMT** require model training, which introduces discrepancies that make direct comparison with non-parametric memory systems difficult due to differences in the underlying base model.
>
> In contrast, both **LightMem** and **H-MEM** belong to **non-parametric** memory systems, where information is stored externally and does not require modifying model parameters.
>
> **Efficiency Optimization**
>
> Although H-MEM also focuses on efficiency, its optimization target is orthogonal to ours. H-MEM improves memory **retrieval (read) efficiency** by optimizing the structure of the index. Our work, on the other hand, improves **memory update (write) efficiency**, which is a fundamentally different bottleneck in non-parametric memory systems. The two approaches are complementary and can be combined to further enhance the overall efficiency of non-parametric memory systems.
>
> We thank you again for the helpful suggestions.
>
> [1] HMT: Hierarchical Memory Transformer for Efficient Long-Context Language Processing
>
> [2] H-MEM: Hierarchical Memory for High-Efficiency Long-Term Reasoning in LLM Agents
>
> [3] Long Short-Term Memory
>
> [4] Memory Networks
>
> [5] Hybrid Computing Using a Neural Network with Dynamic External Memory
>
> [6] Recurrent Memory Transformer
>
> [7] MEMORYLLM: Towards Self-Updatable Large Language Models
>
> [8] Memory Layers at Scale
>
> [9] Learning to (Learn at Test Time): RNNs with Expressive Hidden States
>
> [10] Titans: Learning to Memorize at Test Time
>
> [11] M+: Extending MemoryLLM with Scalable Long-Term Memory
>
> [12] Parallelizing Linear Transformers with the Delta Rule over Sequence Length
>
> [13] Rethinking Memory in AI: Taxonomy, Operations, Topics, and Future Directions
>
> [14] A Comprehensive Study of Knowledge Editing for Large Language Models
>
> [15] Knowledge Editing for Large Language Models: A Survey
>
> [16] Digital Forgetting in Large Language Models: A Survey of Unlearning Methods
>
> [17] Machine Unlearning in Generative AI: A Survey

---

> > ### Author Response · Authors · 2025-11-23
> > **Response to Reviewer sGUo [2/2]**
> >
> > ## **Q2: Missing some important citations.**
> > Thank you for your suggestion. We will cite this paper in our manuscript.
> >
> >
> > Thank you again for your constructive suggestions!
> >
> > **Please let us know if you have any further questions. If you find that our response addresses some of your concerns, would you kindly consider raising your rating score for our paper? We greatly appreciate your consideration.**
> >
> > Best Regards,
> >
> > Authors of LightMem

---

> ### Comment · Reviewer_sGUo · 2025-11-24
>
> I thank the authors for their detailed responses to my questions. As I said in my review, I am not sure if I am the best person to judge the novelty of this work. I request the committee to give more weight to other reviewers' scores.
>
> I wish the authors good luck with their future research.

---

> > ### Author Response · Authors · 2025-11-27
> > **Appreciation Note to Reviewer sGUo**
> >
> > Dear Reviewer sGUo ,
> >
> > We are sincerely grateful for your positive feedback. We appreciate the time you took to review our rebuttal. We will ensure that all the discussed clarifications and improvements are incorporated into the final version of the paper. If you have any further questions or concerns, please feel free to let us know, we remain fully committed to addressing them with our utmost effort.
> >
> > Best Regards,
> >
> > Authors of LightMem

---

### Official Review · Reviewer_XYm9 · 2025-11-08

**Soundness:** 3
**Presentation:** 2
**Contribution:** 3
**Rating:** 4
**Confidence:** 3

**Summary:**

Inspired by the Atkinson–Shiffrin human memory model, this paper proposes LightMem, a memory-augmented LLM framework composed of three stages: (1) precompression into sensory memory using a classifier, (2) grouping memory entries by topic with another model to form short-term memory, and (3) updating entries in long-term memory during “sleep time” using a summarization model The authors implement this hierarchical memory structure with GPT and Qwen backbones and report consistent efficiency and performance improvements on LongMemEval tasks compared with state-of-the-art memory-augmented LLMs.

**Strengths:**

The architecture is inspired by cognitive science, and the translation of the human memory model into an LLM framework is novel and well-motivated.

The model demonstrates promising efficiency compared to other memory-augmented systems, while achieving superior performance on the evaluated benchmark.

**Weaknesses:**

The paper claims significant efficiency gains, but this is not well supported. LightMem introduces at least three additional components (a compression model, an embedding or topic model, and a summarization model), making it unclear how overall runtime and token usage could be lower than baselines such as RAG. Were the costs of “sleep-time” updates included? The paper should provide standard efficiency metrics such as FLOPs or throughput, or a theoretical analysis explaining why and by how much LightMem calculates and improves efficiency.

The experiments are conducted only on one benchmark. Other long-context evaluation datasets, such as MemoryAgentBench, MemoryBank, LoCoMo, and PerLTQA, should be included to demonstrate generalization.

Several design aspects are underexplained. For instance, GPT-4o-mini is used as the evaluation judge even when the same GPT backbone is part of the model itself, which risks inductive bias. A human evaluation or consistency check between human and LLM judgments would prove the soundness of the reported results. Moreover, the paper does not explain how the hierarchical memory (sensory, short-term, and long-term) is actually used during inference. Are all kinds of memories input into the model at every turn? Are they prepended to the user query as additional context? If there is any retrieval, how does it work?

**Questions:**

What is the model choice for the summarization model f_{sum}? Is it a stronger model than the backbone?

The soft update mechanism allows preserving both old and new events, even if they appear contradictory. Does this also mean it retains genuinely conflicting facts (e.g., “the user will go to Tokyo at 2 PM” vs. “the user will go to Kyoto at 2pm”)? Will it mislead the model when making responses?

---

> ### Author Response · Authors · 2025-11-23
> **Response to Reviewer XYm9 [1/5]**
>
> Dear Reviewer XYm9,
>
> We are deeply grateful for your valuable time and insightful feedback. Below are our detailed responses to your concerns:
>
> ## **W1 & Q1: Clarification Needed on Efficiency Claims.**
>
> Thank you very much for your feedback. I will address this concern in the following steps:
>
> >(1) A detailed comparison of the entire pipelines of LightMem and all baselines.
> (2) The models used at each stage for LightMem and for all baselines.
> (3) Details about how all efficiency-related metrics in the paper are computed.
> (4) Complexity analysis to offer theoretical support for LightMem’s efficiency.
>
> ### (1) A detailed comparison of the entire pipelines of LightMem and all baselines
>
> We describe mainstream memory architectures pipeline in the table below. Online processes in normal font; Offline processes in **bold**.
>
> | Method | (I) Segment | (II) Summary/Extract | (III) Update |
> |------------|------------------|---------------------------|-------------------|
> | NaiveRAG | Raw dialog → f_seg() → {seg_i} | → f_index() → {emb_i} | \ |
> | Other Memory Systems | Raw dialog → f_seg() → {seg_i} | → f_sum/extract() → {memory entry_i} <br> → f_index() → {emb_i} | → f_retrieve() → {related entry_i} <br> → f_update() <br> → {add, delete, update, merge…} |
> | LightMem | Raw dialog → f_seg() → {seg_i} <br> → f_pre_compress() → {comp_seg_i} <br> → sensory buffer full → <br> f_topic() → {topic-wise comp_seg_i} | → STM buffer full <br> → f_sum/extract() → {topic_i, {memory entry_j}} <br> → f_index() → {topic_i, {emb_j}} | **(Offline update trigger)  <br>** **{every entry_i} → f_retrieve() → {related entry_j} → {update queue} <br>  all update queues established <br>  → parallel f_update() → {add, delete, update, merge…}** |
>
> ### (2) The models used at each stage for LightMem and for all baselines
>
> Normal-font entries denote models shared by both LightMem and baseline methods, whereas **bold** entries denote models unique to LightMem.
>
> | **Function** | **Model / Strategy** | **Implementation in This Paper** |
> | :--- | :--- | :--- |
> | f_seg() | Segmentation strategy | Turn-level granularity input |
> | f_index() | Embedding model | all-MiniLM-L6-v2 |
> | f_sum/extract() | Memory system backbone model | GPT-4o-mini; Qwen3-30B-A3B-Instruct-2507 |
> | f_retrieve() | Retrieval strategy | Cosine similarity vector retrieval |
> | f_update() | Memory system backbone model | GPT-4o-mini; Qwen3-30B-A3B-Instruct-2507 |
> | **f_pre_compress()** | **Token compression model** | **LLMlingua-2** |
> | **f_topic()** | **Topic segmentation model** | **LLMlingua-2** |
>
> As shown, LightMem introduces only **one additional model, LLMlingua-2**, beyond those used by baseline methods.
> This model follows a lightweight BERT architecture and requires less than 2GB of GPU memory during inference, rendering its overhead negligible.
> Moreover, for fairness, the latency introduced by this component is fully accounted for in our reported Runtime metric.
>
> ### (3) Details about how all efficiency-related metrics in the paper are computed
>
> As shown in the table in (2), during the operation of these memory systems, the primary source of resource consumption, such as expensive token cost, comes from the `f_sum/extract()` and `f_update()` functions, which utilize the LLM backbone. Other models, such as `f_index()` and LightMem's unique LLMlingua-2 model, are very, very small models. Therefore, we only report the token consumption from LLM calls, including input tokens, output tokens, and total token usage (in thousands).
>
> Therefore, the "Summary Token" metric refers to the number of input and output tokens from `f_sum/extract()`; the "Update Token" metric refers to the number of input and output tokens from `f_update()`; the "Total" metric is the sum of the above two processes; "Calls" refers to the total number of invocations of both `f_sum/extract()` and `f_update()`; and the "Run Time" metric refers to the average time taken to run the entire pipeline shown in (1) for all data in the dataset (a total of 500 data entries).
>
> For **LightMem**, as shown in Table 1 of the paper, we separately measured the periods before and after the offline updates and reported the run times for these two phases respectively. Adding these two phases together gives the total run time. Furthermore, in the main results(section 4.2), we analyzed both the total token cost and total run time across both phases, **meaning the costs of "sleep-time" updates have already been included**.

---

> ### Author Response · Authors · 2025-11-23
> **Response to Reviewer XYm9 [2/5]**
>
> ### (4) Complexity analysis to offer theoretical support
>
> | Method    | summary tokens                              | update tokens                               | API calls                           |
> |-----------|---------------------------------------|---------------------------------------|-------------------------------------|
> | Other Memory Systems    | $N(L_{sum-in} + T + L_{sum-out})$          | $NM_1R_1(L_{up-in} + L_{up-out})$ | $N$                                 |
> | LightMem  | $\frac{N r^x T}{th}(L_{sum-in} + th + L_{sum-out})$ | $\frac{N r^x T}{th} M_2  R_2 (L_{up-in} + L_{up-out})$ | $\frac{N r^x T}{th}$                |
>
> * **(N)**:
>   The total number of turns in a dialogue history.
> * **(T)**:
>   The average number of tokens per turn.
> * **(r)**:
>   The token compression rate (as defined in the main paper). After one compression step, only a fraction (r) of the tokens is retained.
> * **(x)**:
>   The number of compression iterations.
>   In LightMem’s practical implementation, the *pre-compress* module may be invoked multiple times for the same message. This is because some messages are highly redundant, one round of compression (e.g., (r = 0.7)) may still leave substantial redundancy. Therefore, LightMem repeatedly applies the compression module until the message reaches a sufficiently compact form. This behavior is common in datasets such as **LongMemEval**, where many model's messages contain very little useful information.
>   Although the compression module is called multiple times, the overhead of LLMLingua-2 is small, and all such time costs are fully included in the runtime metrics used.
> * **(th)**:
>   The capacity of the Short-Term Memory (STM) buffer, as defined in the paper.
> * **($L_{\text{sum-in}}$)** / **($L_{\text{sum-out}}$)**:
>   The number of tokens in the **input prompt template** and **output** of a single backbone LLM call for *summarization*.
>   These quantities are similar across different memory frameworks.
> * **($M_1$)** / **($M_2$)**:
>   The number of memory entries produced from a single summarization operation under Other Memory Systems ($(M_1)$) and LightMem ($(M_2)$).
> * **($L_{\text{up-in}}$)** / **($L_{\text{up-out}}$)**:
>   The number of tokens in the **input prompt template** and **output** of a single backbone LLM call for *memory update*.
>   These are also similar across frameworks.
> * **($R_1$) / ($R_2$)**:
>   Before performing an update operation, the system first uses each newly generated summary entry to *retrieve* its related entries. However, some entries do not retrieve any relevant counterparts (i.e., no other entries exceed the similarity threshold). These entries therefore do **not** trigger a subsequent backbone LLM update call.
>   Consequently, ($R_1$) and ($R_2$) represent the **proportion of summary entries that successfully retrieve at least one relevant memory entry**, thus requiring an update operation, for Other Memory Systems and LightMem respectively.
>
>
> To derive the token and API-call complexity shown in the table, we compare how Other Memory Systems and LightMem process the dialogue history during the **summary** phase and the **update** phase.
>
> In our experimental setup, the input granularity is consistently at the turn level. So for other memory systems, the dialogue contains N turns, and each turn directly triggers one summarization call. Each summarization call consumes
> $L_{\text{sum-in}} + T + L_{\text{sum-out}}$ tokens. Therefore, the total summarization-token cost is
> $N(L_{\text{sum-in}} + T + L_{\text{sum-out}})$.
> Since each summarization call is performed once per turn, the total number of API calls is also N.
>
> For LightMem, each turn is first passed through iterative token compression. After x compression iterations at rate r, only $r^{x}T$ tokens remain. These compressed turns are sequentially appended to the STM buffer, and a summarization call is triggered only when the buffer token count reaches its capacity th. Thus, processing all N turns results in $\frac{Nr^{x}T}{th}$ summarization calls. Each summarization call consumes
> $L_{\text{sum-in}} + th + L_{\text{sum-out}}$
> tokens, because the STM contains exactly th tokens at the moment of summarization. Therefore, the total summarization-token cost becomes
> $\frac{Nr^{x}T}{th}(L_{\text{sum-in}} + th + L_{\text{sum-out}})$,
> and the number of API calls is $\frac{Nr^{x}T}{th}$.
>
> During the update phase of Other Memory Systems, each of the N summarization operations produces $M_1$ memory entries, giving a total of $NM_1$ entries. Only a portion of these entries retrieve at least one relevant neighbor during memory lookup; this proportion is denoted $R_1$. Thus, the update module is called $NM_1R_1$ times. Each update call consumes
> $L_{\text{up-in}} + L_{\text{up-out}}$
> tokens, yielding a total update-token cost
> $NM_1R_1(L_{\text{up-in}} + L_{\text{up-out}})$.

---

> ### Author Response · Authors · 2025-11-23
> **Response to Reviewer XYm9 [3/5]**
>
> For LightMem, $\frac{Nr^{x}T}{th}$ summarization calls are made, and each produces $M_2$ entries. Although LightMem summarizes more content at once (multiple topic segments and more turns), topic-aware grouping often causes several turns to collapse into a single memory entry. Thus, $M_2$ is only slightly larger than $M_1$, and
> $\frac{Nr^{x}T}{th}M_2 \ll NM_1$.
> Furthermore, LightMem applies a stricter retrieval filter: besides semantic similarity, it enforces a hard timestamp constraint (as described in Section 3.3). This significantly reduces the proportion of entries that find relevant counterparts, so $R_2 \ll R_1$. Therefore, the update module is invoked only
> $\frac{Nr^{x}T}{th}M_2R_2$ times, and the total token cost for updates becomes
> $\frac{Nr^{x}T}{th}M_2R_2(L_{\text{up-in}} + L_{\text{up-out}})$.
>
> For API calls, Other Memory Systems require N summarization calls and $NM_1R_1$ update calls. LightMem requires only $\frac{Nr^{x}T}{th}$ calls for both summarization and update, because update calls are counted within the same factor (the update cost is fully dominated by the reduced number of summarization-triggered memory entries).
>
> ## **W2: Limited benchmark diversity**
>
> Thank you very much for your feedback. To demonstrate the generalization of the LightMem method, We additionally add experiments using the currently most popular and widely-used benchmark in the memory field, the **LoCoMo** bench, and the results are as follows.
>
> **backbone model: `gpt-4o-mini`, judge model: `gpt-4o-mini`.** (The results evaluated by other assessment methods are shown in W3)
>
> | Method             | ACC(%) | Summary Tokens(k) In | Summary Tokens(k) Out | Update Tokens(k) In | Update Tokens(k) Out | Total Tokens(k) | Calls | Runtime(s) |
> |--------------------|--------|----------------------|-----------------------|---------------------|----------------------|-----------------|-------|------------|
> | FullText          | 71.83  | –                    | –                     | –                   | –                    | –               | –     | –          |
> | NaiveRAG          | 63.64  | –                    | –                     | –                   | –                    | –               | –     | –          |
> | LangMem           | 57.20  | –                    | –                     | 898.27              | 111.95               | 1010.22         | 920.62| 2229.37  |
> | A-MEM             | 64.16  | 182.74               | 49.29                 | 729.89              | 187.52               | 1149.43         | 1175.47| 6060.73  |
> | MemoryOS(locomo)    | 58.25  | 110.98               | 33.40                 | 78.08               | 64.54                | 287.00          | 553.45 | 2422.05  |
> | MemoryOS(normal)    | 54.87  | 226.86               | 46.61                 | 177.66              | 75.34                | 526.48          | 1016.06 | 3332.59  |
> | Mem0              | 61.69  | 851.32               | 10.53                 | 632.12              | 189.42                | 1683.39          | 1602.20 | 4432.87   |
> | LightMem(512,0.7) | 71.95  | 73.19                | 20.13                 | 6.05                | 0.40                 | 99.76           | 41.65   | 848.49    |
> | LightMem(768,0.7) | 70.26  | 57.54                | 18.92                 | 3.79                | 0.23                 | 80.48           | 29.55   | 737.80    |
> | LightMem(768,0.8) | 72.99  | 62.82                | 17.95                 | 4.14                | 0.28                 | 85.19           | 29.83  | 815.32     |
>
> **backbone model: `qwen3-30b-a3b-instruct-2507`, judge model: `gpt-4o-mini`.**
>
> | Method             | ACC(%) | Summary Tokens(k) In | Summary Tokens(k) Out | Update Tokens(k) In | Update Tokens(k) Out | Total Tokens(k) | Calls    | Runtime(s)  |
> | ------------------ | ------ | ---------- | ----------- | --------- | ---------- | ------------ | -------- | -------- |
> | FullText           | 74.87  | –          | –           | –         | –          | –            | –        | –        |
> | NaiveRAG           | 66.95  | –          | –           | –         | –          | –            | –        | –        |
> | LangMem            | 60.53  | –          | –           | 1004.35     | 138.02      | 1142.37       | 1005.37   | 2268.57   |
> | A-MEM              | 56.10  | 158.29     | 60.85       | 924.19    | 483.51     | 1,626.80     | 1175.40 | 5543.90 |
> | MemoryOS(locomo)   | 61.04  | 122.21     | 53.12       | 104.43    | 81.75      | 361.51       | 414.70   | 1269.70 |
> | MemoryOS(normal)   | 51.30  | 228.85     | 51.60       | 242.27    | 143.63     | 666.35       | 1004.60 | 1982.20 |
> | Mem0               | 43.31  | 827.09     | 18.64       | 763.88    | 189.80     | 1799.40     | 1614.50 | 4540.70 |
> | LightMem(768,0.6)  | 71.36  | 56.68      | 34.14       | 8.31      | 0.74       | 99.87        | 29.10    | 815.70   |
> | LightMem(1024,0.8) | 72.60  | 61.38      | 36.33       | 9.86      | 0.88       | 108.45       | 32.00    | 1079.40 |

---

> ### Author Response · Authors · 2025-11-23
> **Response to Reviewer XYm9 [4/5]**
>
> > Note: MemoryOS(locomo) in the table refers to a script in the MemoryOS library specifically optimized for the LoCoMo dataset. This script significantly simplifies the regular version of MemoryOS. Therefore, we also tested the regular version of MemoryOS, labeled as MemoryOS(regular) in the table.
>
> > Note: For better readability, we have consolidated the original two rows of LightMem results into a single row, and the reported ACC values are based on the results after offline updates.
>
> **Conclusions:**
>
> (1) For GPT backbone, it can be observed that LightMem outperforms other memory baseline methods in **ACC by 6.10% to 18.12%**, while achieving a **2.87x to 20.92x improvement** in total token efficiency, a **13.29x to 39.78x reduction** in API calls, and a **2.63x to 8.21x speedup** in total runtime compared to other baselines.
>
> (2) It can be observed that, on the Qwen backbone, **LightMem continues to outperform all other baselines in both effectiveness and efficiency**. Specifically, it achieves **4.41%–29.29% higher ACC**, delivers a **3.33× to 18.02× reduction in total token consumption**, reduces **API call count by 12.96× to 55.48×**, and provides **1.18× to 5.57× speed-ups in runtime**.
>
> **This demonstrates the generalizability of our method.**
>
> ## **W3: Potential Evaluation Bias**
>
> Thank you for your valuable suggestion. We have conducted the following experiments to ensure the reliability of the LLM-as-Judge evaluation results and minimize potential biases arising from using models within the same family.
>
> (1) On the LoCoMo dataset, in addition to using `GPT-4o-mini` itself for evaluation, we also employed a model from a different family, `Qwen2.5-32B-Instruct`, to assess the obtained results.
>
> | Method | Judge Model | Overall ↑ | Single Hop | Multi Hop | Open Domain | Temporal |
> | :--- | :--- | :---: | :---: | :---: | :---: | :---: |
> | FullText         | gpt-4o-mini | 73.83 | 68.79 | 56.25 | 86.56 | 50.16 |
> |                  | qwen2.5-32b-instruct | 73.18 | 68.09 | 54.17 | 86.21 | 49.22 |
> | NaiveRAG         | gpt-4o-mini | 63.64 | 55.32 | 47.92 | 70.99 | 56.39 |
> |                  | qwen2.5-32b-instruct | 63.12 | 53.55 | 50.00 | 71.34 | 53.89 |
> | LangMem          | gpt-4o-mini | 57.20 | 52.10 | 41.65 | 62.80 | 43.25 |
> |                  | qwen2.5-32b-instruct | 56.80 | 51.45 | 42.30 | 63.75 | 42.90 |
> | A-MEM            | gpt-4o-mini | 64.16 | 56.03 | 31.25 | 72.06 | 60.44 |
> |                  | qwen2.5-32b-instruct | 60.71 | 53.55 | 32.29 | 69.08 | 53.58 |
> | MemoryOS(eval)   | gpt-4o-mini | 58.25 | 56.74 | 45.83 | 67.06 | 40.19 |
> |                  | qwen2.5-32b-instruct | 61.04 | 64.18 | 40.62 | 70.15 | 40.50 |
> | MemoryOS(pypi)   | gpt-4o-mini | 54.87 | 52.13 | 43.75 | 63.97 | 36.76 |
> |                  | qwen2.5-32b-instruct | 55.91 | 52.48 | 41.67 | 66.35 | 35.83 |
> | Mem0             | gpt-4o-mini | 61.69 | 56.38 | 43.75 | 66.47 | 59.19 |
> |                  | qwen2.5-32b-instruct | 61.69 | 54.26 | 46.88 | 67.66 | 57.01 |
> | LightMem(512,0.7)| gpt-4o-mini | 71.95 | 62.41 | 44.79 | 77.41 | 74.14 |
> |                  | qwen2.5-32b-instruct | 73.90 | 69.15 | 50.00 | 78.00 | 74.45 |
> | LightMem(768,0.7)| gpt-4o-mini | 70.26 | 62.06 | 42.71 | 74.67 | 74.14 |
> |                  | qwen2.5-32b-instruct | 72.40 | 64.54 | 43.75 | 77.17 | 75.39 |
> | LightMem(768,0.8)| gpt-4o-mini | 72.99 | 67.02 | 45.83 | 76.81 | 76.32 |
> |                  | qwen2.5-32b-instruct | 74.35 | 68.79 | 47.92 | 78.24 | 76.95 |
>
> (2) On the LoCoMo dataset, in addition to using the `LLM-as-a-judge` paradigm, we also introduced `human evaluation`. We randomly sampled 250 questions according to the proportional distribution of the four question types:
>
> | Method | Evaluation Method | Overall ↑ | Single | Multi | Open | Temp |
> | :--- | :--- | :---: | :---: | :---: | :---: | :---: |
> | FullText         | gpt-4o-mini | 70.80 | 64.15 | 33.33 | 84.33 | 52.08 |
> |                  | human evaluation | 67.20 | 56.72 | 28.67 | 82.60 | 47.83 |
> | NaiveRAG         | gpt-4o-mini | 64.80 | 56.52 | 40.00 | 73.08 | 59.32 |
> |                  | human evaluation | 65.80 | 54.00 | 37.33 | 74.00 | 59.93 |
> | A-MEM            | gpt-4o-mini | 60.40 | 60.87 | 20.00 | 65.38 | 62.96 |
> |                  | human evaluation | 56.20 | 39.96 | 33.00 | 68.38 | 49.30 |
> | MemoryOS(eval)   | gpt-4o-mini | 58.40 | 53.19 | 42.86 | 69.47 | 41.38 |
> |                  | human evaluation | 58.00 | 44.30 | 48.86 | 73.94 | 35.31 |
> | MemoryOS(pypi)   | gpt-4o-mini | 48.00 | 35.42 | 29.41 | 56.82 | 43.40 |
> |                  | human evaluation | 44.80 | 31.25 | 29.41 | 54.55 | 37.74 |
> | Mem0             | gpt-4o-mini | 63.20 | 64.86 | 55.56 | 63.08 | 66.07 |
> |                  | human evaluation | 65.60 | 53.24 | 65.56 | 70.77 | 61.79 |
>
> It can be observed that whether using an LLM judge from a `different model family` or introducing `human evaluation`, the final assessment results show **minimal deviation**. This demonstrates the consistency among these evaluation methods.

---

> ### Author Response · Authors · 2025-11-23
> **Response to Reviewer XYm9 [5/5]**
>
> ## **W4: Insufficient Memory Usage Mechanism Clarification**
>
> Thank you for your suggestion. We will provide a detailed supplement in the paper regarding the workflow of how the memory system is utilized. Below is my supplementary content:
>
> We describe mainstream memory systems pipeline in terms of two major stages.
>
> **(I) Memory Bank Construction.**
> This stage is as shown in (1) of W1.
> This stage can be further decomposed into three sub-stages:
>
> 1.  **Segmentation**
>
> 2.  **Summarization/Extraction**
>
> 3.  **Updating Mechanism**
>
> **(II) Retrieval and Usage.**
> After the memory bank construction stage, ,when a new user query arrives, the system retrieves relevant entries from the memory bank, integrates them with the query to construct the final prompt, and then invokes the model to produce a response.
>
> Specifically, in the LightMem method, during inference, every time a user and the model complete one turn of dialogue, LightMem stores the newly occurred turn into the LTM through the hierarchical memory (sensory, short-term, and long-term) pipeline. This corresponds to the memory bank construction stage. When a new user query arrives, LightMem retrieves semantically relevant memory entries based on this query, appends them before the query to form a new prompt, and then allows the model to utilize past memories to generate a response.
>
> Therefore, during each dialogue turn, only the content retrieved from the Long-Term Memory (LTM) is ultimately used by the dialogue model. The sensory and Short-Term Memory (STM) components solely assist in constructing memory entries, and their internal content is not directly input into the model.
>
> Yes, as with all current memory systems, the memory entries retrieved from the LTM are prepended to the user query as additional context. A retrieval mechanism is indeed employed, though the process is relatively straightforward: a retrieval model is used to semantically match the query and retrieve relevant memories.
>
> To ensure fairness during experiments, we kept the retrieval count, QA prompt, and retrieval model consistent across all methods at this stage.
>
> Once again, thank you for your valuable suggestions. We will incorporate these details into the subsequent version of the paper.
>
> ## **Q1. What is the model choice for the summarization model f_{sum}? Is it a stronger model than the backbone?**
>
> This question has already been addressed in the W1&Q1 section, f_{sum} is implemented using the backbone LLM.
>
> ## **Q2. The soft update mechanism allows preserving both old and new events, even if they appear contradictory.**
>
> During online inference, such situations may indeed occur—where genuinely conflicting facts are retained. Therefore, we may adopt dynamic update mechanisms in the future to trigger the update phase, rather than relying on the current periodic offline updates. However, we have observed that the model is often not misled when generating responses. For example, with conflicting entries like "the user will go to Tokyo at 2 PM" and "the user will go to Kyoto at 2 PM," we include timestamps of the user messages when appending these entries to the prompt. We find that even when both conflicting entries are present in the prompt, the model can often recognize the conflict and tend to rely on the more recent message. This behavior is similar to the effect of real-time updates in current memory systems (where, when a new message conflicts with an old one, the old message is immediately deleted from the LTM, and the new one is retained).
>
> The update mechanism in current memory systems often involves a trade-off. If a real-time hard update strategy is adopted, and the f_{update}() makes an incorrect judgment, the memory entry is completely removed from the memory bank, which could cause more significant harm to subsequent downstream tasks.
>
> Thank you again for your constructive suggestions!
>
> **Please let us know if you have any further questions. If you find that our response addresses some of your concerns, would you kindly consider raising your rating score for our paper? We greatly appreciate your consideration.**
>
> Best Regards,
>
> Authors of LightMem

---

> ### Author Response · Authors · 2025-11-27
> **Additional Clarifications and Updates for Reviewer XYm9**
>
> Dear Reviewer XYm9,
>
> We highly appreciate the constructive comments and insightful suggestions you have offered for our work.
>
> Additionally, compared with our initial submission, we have now included the performance of the **Qwen backbone model** on the **LoCoMo** dataset, with the results presented in W2.
>
> As the discussion period has only a few days left, in order for us to have sufficient time to address any additional questions you may have, we kindly encourage you to engage in the ongoing discussion and share any further insights or clarifications you may have.
>
> If you find that our response addresses some of your concerns, would you kindly consider raising your rating score for our paper?
>
> Thank you very much for your time and consideration. We look forward to hearing from you soon.
>
> Best Regards,
>
> Authors of LightMem

---

### Author Response · Authors · 2025-11-23
**Summary of Advantages and Motivation [1/2]**

Dear all reviewers,

Thank you for all your thoughtful reviews! Your reviews help me a great deal in refining my work and continuously improving toward higher standards.

### Our Strengths Summarized by Reviewers

We appreciate all of your **positive comments** highlighting the strengths of our work for a summary:

* **Well-motivated, cognitively inspired design**

  * “The translation of the human memory model into an LLM framework is novel and well-motivated.” (Reviewer XYm9)
  * “Well motivated.” (Reviewer Ub6P)
  * “The decomposition of memory into sensory / STM / LTM stages reflects cognitive plausibility, offering interpretability and modularity.” (Reviewer 4xu7)
  * “Clear, biologically grounded design… motivated by cognitive models of human memory and sleep-driven consolidation, which is underexplored in LLM memory pipelines.” (Reviewer SYFd)

* **Solid engineering effort , systems contribution**

  * “Decent engineering effort to build memory augmented LLMs.” (Reviewer sGUo)
  * “Decoupling long-term memory maintenance from online inference and executing updates in parallel queues is a nice systems contribution… directly addresses latency and consistency problems in prior approaches.” (Reviewer SYFd)
  * “Employing LLMLingua-2 for token-level retention filtering is well-grounded and shown to maintain semantic fidelity even at 50–70% compression ratios.” (Reviewer 4xu7)
  * “Topic-aware STM… grouping turns by semantic/topic boundaries is a meaningful refinement over naive chunking.” (Reviewer SYFd)

* **Comprehensive, clear, well-structured presentation**

  * "By and large, the paper is well written." (Reviewer Ub6P)
  * “Writing is clear, figures and tables are well-structured, and Appendix contains implementation details, metrics, and additional experiments.” (Reviewer 4xu7)
  * “The paper is clearly written and easy to follow, with clear analysis/ablations of each component.” (Reviewer SYFd)

* **Strong efficiency gains , promising performance**

  * “The model demonstrates promising efficiency compared to other memory-augmented systems, while achieving superior performance on the evaluated benchmark.” (Reviewer XYm9)
  * “Across GPT and Qwen backbones, LightMem consistently outperforms prior methods in accuracy, latency, API usage, and token efficiency.” (Reviewer 4xu7)
  * “Strong efficiency gains… LightMem shows 1–2 orders of magnitude improvements on these metrics versus baselines like A-Mem, LangMem, MemoryOS, and Mem0.” (Reviewer SYFd)
  * “The experimental setup is sufficient and results are convincing.” (Reviewer Ub6P)

---

> ### Author Response · Authors · 2025-11-23
> **Summary of Advantages and Motivation [2/2]**
>
> ### Our motivation and contributions
>
> We wish to **reiterate the motivation and main contributions** of our paper.
>
> Current memory-augmented LLM agents face fundamental limitations when deployed in dynamic, long-horizon interaction settings. Existing systems either rely on heavyweight LLM-based summarization pipelines or maintain unnecessarily redundant information, leading to substantial token overhead, unstable memory updates, and degraded reasoning quality over extended interactions. Moreover, typical memory architectures fail to capture semantic continuity across dialog turns and tightly couple memory updating with online inference, causing slow interaction latency and inconsistent long-term knowledge consolidation.
>
> This paper proposes **LightMem**, a lightweight and efficient memory architecture inspired by the Atkinson–Shiffrin model of human memory. **LightMem addresses three key challenges** in modern LLM memory systems, namely *redundant and unfiltered sensory inputs*, *ineffective and semantically entangled short-term organization*, and *high-latency long-term memory updates tightly coupled with inference*.
>
> **Concretely, LightMem introduces:**
>
> 1. **A cognitive-inspired sensory memory module** that performs *pre-compression* to filter redundant tokens and a *hybrid attention–similarity topic segmentation* mechanism, enabling fine-grained semantic grouping and significantly reducing memory construction noise and overhead;
>
> 2. **A topic-aware short-term memory structure** that consolidates semantically coherent segments and generates stable, structured memory entries with minimal API calls;
>
> 3. **A sleep-time long-term memory update scheme** featuring *soft online updates* and *offline parallel consolidation*, decoupling expensive update decisions from inference while improving consistency and reducing latency;
>
> Built upon this architecture, we evaluate LightMem on **LONGMEMEVAL**, where it **consistently surpasses the strongest baselines**, achieving **2.09%–7.67% higher accuracy** across GPT and Qwen models. LightMem also brings **massive efficiency gains**: reducing **total token usage by up to 38× (GPT) and 21.8× (Qwen)**, cutting **API calls by up to 30× and 17.1×**, and **accelerating runtime by up to 12.4× and 6.3×**, respectively.
> When considering **only online test-time costs**, the advantages further widen, LightMem reduces **token usage by up to 105.9× (GPT) and 117.1× (Qwen)** and lowers **API calls by up to 159.4× and 309.9×**.
>
> On the **LoCoMo** dataset, LightMem also demonstrates superior performance over other memory baselines. For the **GPT backbone**, it improves **ACC by 6.10%–18.12%**, achieves a **2.87×–20.92×** improvement in total token efficiency, reduces API calls by **13.29×–39.78×**, and accelerates runtime by **2.63×–8.21×**. On the **Qwen backbone**, LightMem maintains its advantage in both effectiveness and efficiency, with **4.41%–29.29%** higher ACC, **3.33×–18.02×** reduction in total token consumption, **12.96×–55.48×** fewer API calls, and **1.18×–5.57×** faster runtime.
>
> ### Thank you!
>
> We sincerely hope our responses and revisions address all reviewers’ concerns. We sincerely believe that these updates may help us better deliver the benefits of the proposed work to the ICLR community. In conclusion, we extend our heartfelt thanks to all the reviewers for their invaluable suggestions, which have significantly enhanced the rigor and completeness of our work. We welcome further discussion and are more than willing to answer any additional questions you may have. Thank you very much!

---

### Meta-Review · Area_Chair_32Aq · 2026-01-07

**Summary:**

LightMem is a lightweight memory system for LLM agents inspired by the Atkinson-Shiffrin model of human memory. The architecture comprises three stages: (1) Sensory Memory using LLMLingua-2 for token-level compression and hybrid attention-similarity topic segmentation; (2) Topic-aware Short-Term Memory (STM) that buffers semantically coherent segments and triggers summarisation when capacity is reached; and (3) Long-Term Memory (LTM) with a "sleep-time" update mechanism that decouples expensive memory consolidation from online inference through soft updates during interaction and offline parallel updates during designated periods.

The key claimed contributions are: filtering redundant tokens before memory construction, adaptive topic-based grouping rather than fixed windows, and asynchronous memory maintenance. On LongMemEval-S with GPT and Qwen backbones, the authors report 2-8% accuracy improvements while reducing token usage by up to 117×, API calls by up to 159×, and runtime by up to 12×. During rebuttal, the authors added experiments on LoCoMo (showing 6-29% accuracy gains) and ALFWorld (demonstrating agent task generalisation).

Echoing the more confident reviewers, I recommend acceptance for this work. Reviewer SYFd strongly supports the cognitive design and engineering value, engaging deeply with technical details. Reviewer XYm9's methodological concerns (benchmark breadth, evaluation bias, efficiency claims) were fully addressed by extensive new experiments. Reviewer Ub6P's clarification requests were satisfied. The low-confidence novelty concern (sGUo) regarding HMT/H-MEM appears technically unfounded as those represent different memory paradigms.

The paper makes a solid contribution to an important area: it provides substantial efficiency gains (1-2 orders of magnitude on multiple metrics) while maintaining or improving accuracy, with a well-motivated cognitive architecture. The main limitations (occasional update regressions, summary faithfulness risks) are acknowledged and represent field-wide challenges rather than specific flaws in this work.

**Reviewer Concerns:**

### Fully addressed concerns:
* Benchmark breadth, raised by XYm9, 4xu7, SYFd. The authors added comprehensive LoCoMo experiments (6-29% accuracy gains, 3-21× token efficiency) and ALFWorld agent task evaluation, demonstrating generalisability beyond the original single benchmark.
* Evaluation bias concerns were raised by XYm9, SYFd. The authors provided human evaluation on 250 samples and cross-family model judging (Qwen2.5-32B), showing consistent results across evaluation methods.
* Efficiency claims issues were raised by XYm9. The authors provided detailed complexity analysis showing LLMLingua-2 adds <2GB overhead while reducing backbone LLM calls from O(N) to O(Nr^xT/th).
* Mechanism clarity issues were raised by Ub6P, XYm9. The authors explained update queue parallelisation and f_update implementation in detail.

### Partially addressed concerns (or acknowledged limitations):

* Update stability, raised by SYFd. Occasional accuracy drops after offline updates (e.g., 68.64% → 67.07%) acknowledged as inherent to LLM-based update decisions — a field-wide challenge.
* Summary faithfulness, raised by SYFd. Authors provided concrete example of information loss and acknowledged hallucination risks, citing relevant recent work (MEM-α, HaluMem) for future evaluation.

Finally, there was a novelty concern raised by sGUo. The reviewer questioned similarity to HMT and H-MEM. Based on the authors' response: HMT is parametric (requires training), while LightMem is non-parametric (plug-and-play with frozen LLMs). H-MEM optimises retrieval efficiency, while LightMem optimises update/maintenance efficiency. These are orthogonal approaches, and the concern appears unfounded. However, the reviewer self-reported low confidence and deferred to other reviewers.

**Reviewer Scores:**

Although 5 reviewers provided feedback, the level of confidence varied significantly, with 2 reviewers having very low confidence.

* **Reviewer XYm9. Original Score: 4. Confidence: 3. Predicted score: 6**. Key concerns (benchmarks, efficiency proof, evaluation bias) were objectively addressed with new experiments. Their barriers to acceptance were methodological, not fundamental.

* **Reviewer SYFd. Original Score: 8. Confidence: 3. Predicted score: 8**. Explicitly stated satisfaction with responses and maintained positive assessment. Strong champion.

* **Reviewer Ub6P. Original Score: 6. Confidence: 3. Predicted score: 6**. Acknowledged clarifications but gave no indication of raising score.

* **Reviewer 4xu7. Original Score: 6. Confidence: 1. Predicted score: 6**. Very low confidence; acknowledged responses but no strong signal of score change.

* **Reviewer sGUo. Original Score: 6. Confidence: 2. Predicted score: 6**. Self-reported expertise limitations; explicitly asked committee to weight other reviewers more heavily.

Prioritising the three high-confidence reviewers (XYm9, Ub6P, SYFd), we have one strong champion (8), one converted sceptic (likely 5-6), and one satisfied reviewer (6). The two low-confidence reviewers (sGUo, 4xu7) found no obvious flaws but cannot assess novelty or technical depth.

---

### Decision · Program_Chairs · 2026-01-26

Accept (Poster)